# A long-acting interleukin-7, rhIL-7-hyFc, enhances CAR T cell expansion, persistence, and anti-tumor activity

Miriam Y. Kim[1], Reyka Jayasinghe[1], Jessica M. Devenport[1], Julie K. Ritchey[1], Michael P. Rettig [1], Julie O'Neal [1], Karl W. Staser[1,2], Krista M. Kennerly[1], Alun J. Carter[1], Feng Gao[3], Byung Ha Lee [4], Matthew L. Cooper[1,5] & John F. DiPersio [1,5✉]

Chimeric antigen receptor (CAR) T cell therapy is routinely used to treat patients with refractory hematologic malignancies. However, a significant proportion of patients experience suboptimal CAR T cell cytotoxicity and persistence that can permit tumor cell escape and disease relapse. Here we show that a prototype pro-lymphoid growth factor is able to enhance CAR T cell efficacy. We demonstrate that a long-acting form of recombinant human interleukin-7 (IL-7) fused with hybrid Fc (rhIL-7-hyFc) promotes proliferation, persistence and cytotoxicity of human CAR T cells in xenogeneic mouse models, and murine CAR T cells in syngeneic mouse models, resulting in long-term tumor-free survival. Thus, rhIL-7-hyFc represents a tunable clinic-ready adjuvant for improving suboptimal CAR T cell activity.

[1] Division of Oncology, Department of Medicine, Washington University in St. Louis, St. Louis, MO, USA. [2] Division of Dermatology, Department of Medicine, Washington University in St. Louis, St. Louis, MO, USA. [3] Division of Public Health Sciences, Department of Surgery, Washington University School of Medicine, St. Louis, MO, USA. [4] NeoImmuneTech, Inc., Rockville, MD, USA. [5] These authors jointly supervised this work: Matthew L. Cooper, John F. DiPersio. ✉email: jdipersi@wustl.edu

Chimeric antigen receptor (CAR) T cell therapy is transforming modern cancer therapy, with four anti-CD19 CAR T cell products Food and Drug Administration-approved for relapsed/refractory B cell malignancies[1–4], and many other CAR T cells for a variety of tumors currently being tested in clinical trials. However, in clinical practice suboptimal CAR T cell efficacy often results in antigen-positive tumor escape and disease relapse[5,6]. As the expansion and persistence of CAR T cells correlate with clinical response, we reasoned that interleukin-7 (IL-7), a pro-lymphoid growth and survival factor involved in the development, maintenance, and proliferation of T cells, would promote CAR T cell efficacy and tumor cell killing in vivo.

rhIL-7-hyFc (efineptakin alfa, NT-I7) is a homodimeric genetically modified IL-7 molecule fused to a stable hyFc platform[7], which prevents complement activation while prolonging in vivo serum half-life, providing clear pharmacologic advantages to short-lived native recombinant human IL-7 (rhIL-7). In the first human trial with healthy volunteers (NCT02860715), a single-dose intramuscular injection of rhIL-7-hyFc (60 mcg/kg) resulted in a substantial increase in the number of CD4+ and CD8+ T cells, and no major adverse events or dose-limiting toxicities were reported[8]. Therefore, we decided to explore the potential of this reagent to augment the activity of CAR T cells.

Here we show that rhIL-7-hyFc dramatically enhances CAR T cell expansion, persistence, and anti-tumor efficacy, resulting in significantly prolonged survival of mice in CD19+ lymphoma and CD33+ leukemia xenograft models treated with human CAR T cells, and in immunocompetent mice bearing CD19+ tumors treated with murine CAR T cells. Immunocompetent mice that received rhIL-7-hyFc with CAR T cells exhibited sustained B cell aplasia and were protected against tumor rechallenge. Functional studies and single-cell RNA-seq (scRNA-seq) reveal that rhIL-7-hyFc not only increases CAR T cell numbers but also improves T cell cytotoxicity and decreases exhaustion. Therefore, rhIL-7-hyFc has the potential to improve the outcomes of CAR T cell treatment in patients with refractory malignancies.

## Results

**rhIL-7-hyFc enhances the expansion of UCART19 in vitro and maintains CAR-T polyfunctionality.** To test the effects of rhIL-7-hyFc (Fig. 1a) on CAR T cells in vitro, human TCR-deficient anti-CD19 CAR T cells (UCART19) were cultured with rhIL-7-hyFc in the presence of CD19+ tumor cells. We used TCR deleted human CAR T cells to eliminate the potential for xenogeneic graft versus host disease (GVHD) confounding future in vivo experiments. UCART19 was generated from human T cells derived from healthy donors activated with anti-CD3/anti-CD28 beads for two days before electroporation with Cas9 mRNA and a *TRAC*-targeted gRNA. T cells were then transduced with a lentivirus expressing a 3rd generation anti-CD19 CAR (19-28-BBζ) containing a 2A-cleaved human CD34 extracellular domain, permitting ex vivo purification and identification of CAR+ cells (Fig. 1b). CRISPR/Cas9-mediated *TRAC* deletion efficiency was more than 97%, with an average 58% CAR transduction efficiency (Supplemental Fig. 1). Ramos, a CD19+ B-lymphoma cell line, transduced with click beetle red luciferase (CBR) and green fluorescent protein (GFP; hereafter, Ramos^CBR-GFP), was co-cultured with UCART19 at an effector to target (E:T) ratio of 2:1 in either vehicle control, rhIL-7 10 ng/ml, or rhIL-7-hyFc (10, 100 or 1000 ng/ml). Seven days later, UCART19 was replated in fresh media containing rhIL-7 or rhIL-7-hyFc, and re-challenged with Ramos^CBR-GFP (Fig. 1c). Vehicle only controls failed to expand over the 14-day assay period, while

supplementation with either rhIL-7 or rhIL-7-hyFc at concentrations in excess of 10 ng/ml led to robust expansion of UCART19 (Fig. 1d). Qualitative assessment of UCART19 on day 14 post co-culture with rhIL-7 or rhIL-7-hyFc revealed that even at the lowest concentration, rhIL-7-hyFc promoted UCART19 viability, prevented apoptosis, and increased proliferation to the same degree as rhIL-7 (Fig. 1e).

To further characterize rhIL-7-hyFc expanded UCART19 cells, single-cell cytokine analyses using the Isoplexis Isolite platform were performed on UCART19 isolated and purified from vehicle and rhIL-7-hyFc (100 ng/ml) treated samples on day 14 and compared to UCART19 from day 0, prior to tumor challenge. The Isolite Isocode assay enables the detection and quantification of 32 secreted cytokines from individual CAR T cells to define polyfunctional strength index (PSI), a quantitative marker of CAR-T functionality[9]. Despite the robust rhIL-7-hyFc-mediated UCART19 expansion, no loss of functionality was associated with rhIL-7-hyFc treatment as defined by PSI, with the rhIL-7-hyFc expanded UCART19 maintaining a similar PSI to UCART19 input cells from the start of the experiment on day 0 (Fig. 1f and Supplemental Fig. 2A, B). Polyfunctionality was driven by the secretion of effector molecules such as granzyme B, IFNγ, MIP-1α, MIP-1β, perforin, TNF-α, and TNF-β (Supplemental Fig. 2C).

Flow cytometry-based immunophenotyping revealed that samples treated with rhIL-7-hyFc preferentially maintained the percentage of CAR+ cells, while a significant loss of CAR-expressing cells was noted over the duration of the assay in samples treated with vehicle (Fig. 1g). Additionally, rhIL-7-hyFc also increased the proportion of central memory T cells of UCART19 after serial in vitro tumor challenge as compared to vehicle-treated cells (Fig. 1h). Overall, these data indicate that rhIL-7-hyFc potentiates the expansion of functional UCART19 cells in vitro.

**rhIL-7-hyFc enhances UCART19 expansion, persistence, and anti-tumor efficacy in vivo resulting in prolonged survival of CD19+ tumor-bearing NSG mice.** We predicted rhIL-7-hyFc would enhance in vivo expansion of UCART19 and improve anti-tumor efficacy. To test this hypothesis, we used a preclinical mouse model where UCART19 treatment of tumor-bearing NSG mice leads to only a modest increase in survival. NSG mice were injected with $5 \times 10^5$ Ramos^CBR-GFP four days before treatment with $1 \times 10^6$ UCART19 cells, followed by 10 mg/kg of rhIL-7-hyFc subcutaneously on days +1, +15, and +29 after UCART19 infusion (Fig. 2a). Of note, lymphoid tumor cell lines (Ramos, HH, Jurkat, and CCRF-CEM) express the IL-7 receptor (CD127), albeit at levels lower than activated normal T cells (Supplemental Fig. 3), raising the possibility that a preponderant effect of rhIL-7-hyFc on tumor growth would outweigh any beneficial effect of rhIL-7-hyFc on UCART19 activity and tumor killing. However, survival of NSG mice bearing Ramos^CBR-GFP after rhIL-7-hyFc treatment was equivalent to mice receiving Ramos^CBR-GFP cells alone (Fig. 2b), indicating no overtly deleterious pro-lymphoma effects of rhIL-7-hyFc in this model. As expected, Ramos^CBR-GFP bearing NSG mice had only a modest survival benefit after UCART19 treatment alone in this aggressive B lymphoma model compared to untreated mice. Remarkably, 100% of Ramos^CBR-GFP bearing mice treated with UCART19 and rhIL-7-hyFc were alive beyond 80 days, with no clinical signs of xenogeneic GVHD. Bioluminescent imaging (BLI) revealed minimal tumor signal in UCART19 + rhIL-7-hyFc-treated mice over long-term follow-up (Fig. 2c, d). Additionally, quantitative flow cytometric analyses of peripheral blood revealed a four log-fold expansion of UCART19 in rhIL-7-hyFc-treated mice compared to vehicle-treated mice (Fig. 2e, f and Supplemental Fig. 4). Contrary to our in vitro

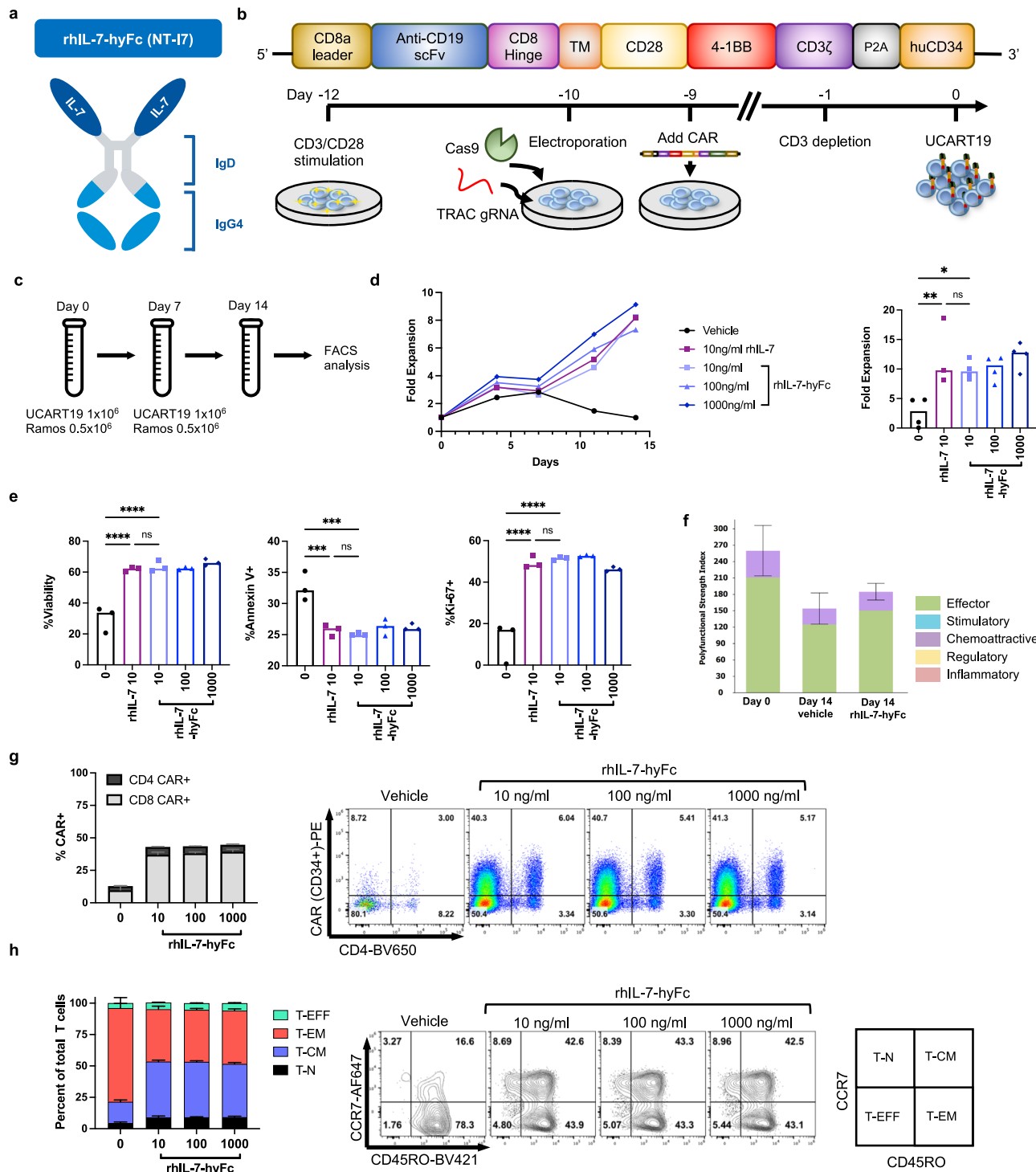

results, the UCART19 cells expanded in vivo with rhIL-7-hyFc demonstrated a more pronounced effector memory phenotype compared to the input T cells (Supplemental Fig. 5).

**rhIL-7-hyFc enhances expansion, persistence, and anti-tumor efficacy of UCART33 resulting in prolonged survival of CD33+ tumor-bearing NSG mice.** To confirm that the effects of rhIL-7-hyFc on CAR T cells were not limited to UCART19 only, we tested rhIL-7-hyFc in a similar way using an orthogonal myeloid-targeting CAR T in an NSG model of acute myeloid leukemia. UCART33 is a second-generation CAR with a 4-1BB, CD3ζ endodomain and an extracellular scFv targeting human

CD33. CAR T cells were generated as previously described for UCART19, with gene editing of *TRAC* to allow for long-term follow-up of NSG mice without xenogeneic GvHD. NSG mice were injected with $5 \times 10^4$ U937[CBR-GFP] four days prior to the administration of $1 \times 10^6$ UCART33 cells, followed by rhIL-7-hyFc 10 mg/kg subcutaneously on days +1, +15, and +29 (Fig. 3a). While UCART33 therapy alone only moderately prolonged median survival in the absence of rhIL-7-hyFc, all mice receiving UCART33 in combination with rhIL-7-hyFc remained alive until the end of the experiment at 175 days post-UCART33 infusion (Fig. 3b). No tumor was detectable in any of the rhIL-7-hyFc treated U937[CBR-GFP] bearing NSG mice after 35 days until

**Fig. 1 rhIL-7-hyFc enhances UCART19 expansion and function in vitro. a** rhIL-7-hyFc is an engineered IL-7 homodimer fused to IgD and IgG4 elements (hyFc©), promoting in vivo stability and reducing complement activation. **b** UCART19 was generated by activating human T cells with anti-CD3/CD28 beads, followed by CRISPR/Cas9 deletion of the human T cell receptor alpha subunit (*TRAC*), and lentiviral transduction of a third-generation anti-CD19 CAR. **c–e** UCART19 was incubated with CD19+ Ramos$^{CBR-GFP}$ at an effector to target (E:T) ratio of 2:1 in 10 ng/ml of rhIL-7 or the indicated concentrations of rhIL-7-hyFc. On day 7, UCART19 was replated with Ramos$^{CBR-GFP}$ at an E:T of 2:1 with the addition of fresh cytokines. UCART19 numbers and phenotypes were determined by serial counting and flow cytometry. Bar graphs represent median values, with each data point representing a different T cell donor. **d** Representative growth plot for a single donor (left) and compiled fold-expansion of all donors on day 14 (right, $n = 3$ donors for rhIL-7, 4 donors for all other groups). *p* Values were calculated by one-way ANOVA with Tukey's multiple comparisons test (rhIL-7 10 vs. 0, $p = 0.0055$, rhIL-7-hyFc 10 vs. 0, $p = 0.0246$, rhIL-7 10 vs. rhIL-7-hyFc, $p = 0.82$). **e** UCART19 viability (%propidium iodide exclusion, rhIL-7 10 vs. 0: $p < 0.0001$; rhIL-7-hyFc 10 vs. 0: $p < 0.0001$; rhIL-7 10 vs. rhIL-7-hyFc: $p = 0.9999$), apoptosis (%annexinV+, rhIL-7 10 vs. 0: $p = 0.0005$; rhIL-7-hyFc 10 vs. 0: $p = 0.0002$; rhIL-7 10 vs. rhIL-7-hyFc: $p = 0.9615$) and proliferation (%Ki-67+, rhIL-7 10 vs. 0: $p < 0.0001$; rhIL-7-hyFc 10 vs. 0: $p < 0.0001$; rhIL-7 10 vs. rhIL-7-hyFc: $p = 0.9805$) measured after rhIL-7 or rhIL-7-hyFc treatment in the presence of target cells after 14 days ($n = 3$ different donors). P-values were calculated by one-way ANOVA with Tukey's multiple comparisons test. **f** UCART19 was incubated with Ramos at an E:T of 1:1 for 16 h with either vehicle or rhIL-7y-hyFc (100 ng/ml) prior to loading purified CAR T cells onto the IsoCode chip for single-cell multiplex cytokine analysis. Polyfunctional strength index (PSI), as defined by the percentage of cells secreting multiple cytokines multiplied by the mean fluorescence intensity of the secreted proteins, was calculated for UCART19 on day 0 (UCART19 input cells), day 14 with vehicle, or day 14 with rhIL-7-hyFc. Bar graphs represent mean ± standard error of the mean ($n = 1$ donor, 2 technical replicates for day 0, 3 technical replicates for day 14 vehicle/rhIL-7-hyFc). Further details regarding the cytokines assayed, polyfunctionality, and signal strength is depicted in Supplemental Fig. 2. **g, h** UCART19 cells serially expanded in the presence of rhIL-7-hyFc and CD19+ Ramos cells at a 2:1 E:T ratio as in **c** were evaluated by flow cytometry on day +14. Bar graphs represent mean ± SD of four technical replicates ($n = 1$ donor). **g** CAR expression analysis on day +14 in CD4+ and CD8+ T cells (left), with representative flow cytometry plots of UCART19 using CD34 to define CAR-T populations (right). **h** Memory phenotype of T cells as measured by expression of CCR7 and CD45RO (T-EFF effector T cells, T-EM effector memory T cells, T-CM central memory T cells, T-N naive T cells). Source data are provided as a Source data file. ns: not significant, $*p \leq 0.05$, $**p \leq 0.01$, $***p \leq 0.001$, $****p \leq 0.0001$.

the end of the experiment (Fig. 3c, d). A logarithmic expansion of UCART33 was seen only in mice that also received rhIL-7-hyFc (Fig. 3e). Of note, UCART33 expansion was not observed with rhIL-7-hyFc in a cohort of mice that did not have prior tumor engrafted, indicating that antigen recognition is also required for rhIL-7-hyFc driven CAR T cell expansion. The dramatic enhancement of UCART33 expansion, persistence, and anti-tumor efficacy against CD33+ AML targets in vivo by rhIL-7-hyFc recapitulate our data generated using UCART19, suggesting that rhIL-7-hyFc potentiation of UCART expansion and clinical efficacy is likely applicable to multiple other CAR T platforms and targets.

**rhIL-7-hyFc can improve CAR T cell expansion and persistence in immunocompetent mice.** The dramatic expansion of CAR T cells in NSG mice with rhIL-7-hyFc treatment prompted concern that rhIL-7-hyFc may exacerbate the toxicities of CAR T cell therapy in human subjects. Alternatively, while rhIL-7-hyFc can support CAR T cell expansion in NSG mice that lack endogenous lymphoid cells, in an immunocompetent host it would also stimulate the endogenous immune system, which may potentially lead to rejection of CAR T cells. To address these concerns, we transitioned to an immunocompetent mouse model, to better evaluate the effects of rhIL-7-hyFc on CAR T cells in the setting of a functional host immune system, and to assess whether this may compromise the health of the host. C57BL/6 mice were injected with syngeneic leukemia cells expressing CD19, followed by treatment with untransduced (UTD) control T cells or murine CD19-targeting CAR T cells (mCART19), then given rhIL-7-hyFc (Fig. 4a). In this experiment, we used high doses of mCART19 ($6 \times 10^6$ cells/mouse) to ensure that any potential toxicity of this therapy would manifest. Tumor cells were derived from a murine acute promyelocytic leukemia cell line, 9523, previously generated in our lab, to which murine CD19 expression was added by lentiviral gene transfer. Adoptively transferred T cells were derived from CD45.1+ donor splenocytes, to facilitate tracking of the cells in CD45.2+ recipient mice, and transduced with a murine CD19-targeting CAR construct that also contained GFP for better tracking of CAR+ cells. UTD cells were expanded simultaneously with mCART19 but not transduced

with the CAR. Prior to T cell infusion mice were given cyclo-phosphamide (Cytoxan) for lymphodepletion, as this has been shown to be important for the activity of adoptively transferred cells[10].

mCART19 alone at this dose was highly effective, and the majority of mice survived long-term, regardless of rhIL-7-hyFc administration (Fig. 4b). Importantly, no toxicity was observed in any of the mice that received mCART19 and rhIL-7-hyFc. Mice did not lose weight or exhibit any clinical signs of distress, and hemoglobin and platelet levels remained for the most part within the normal range (Supplemental Fig. 6a, b). Plasma IL-6 levels were mildly elevated at day +5 after T cell infusion (Supplemental Fig. 6c) and subsequently became undetectable at day +10 for all groups (data not shown). In these mice that had a fully functional immune system, total white blood cell (WBC) counts increased in all mice that received rhIL-7-hyFc, but the degree of elevation did not reach the supra-physiological levels seen in NSG mice (Fig. 4c). Numbers of peripheral blood mCART19, as assessed by flow cytometry for GFP+ cells, increased by approximately 4-fold with rhIL-7-hyFc as compared to mCART19 alone (Fig. 4d and Supplemental Fig. 7). Additionally, in mice that received rhIL-7-hyFc, mCART19 was detected beyond 6 weeks, in contrast to mice receiving mCART19 alone, where minimal CAR T cells were detected after 3 weeks. These findings confirm that rhIL-7-hyFc prolongs CAR T cell expansion and persistence even in an immunocompetent host, without undue toxicity.

Flow cytometry analysis also revealed that the leukocytosis seen after rhIL-7-hyFc treatment was predominantly of recipient origin (CD45.2+), and adoptively transferred CD45.1+ cells comprised only 10-20% of total WBCs across all timepoints (Fig. 4e). Control mice had an increase of both recipient B and T cells with rhIL-7-hyFc, while in mCART19-treated mice recipient T cells were the main drivers of leukocytosis, due to the expected loss of B cells after this treatment (Fig. 4e, f). Notably, recipient T cells outnumbered mCART19 by approximately 10-fold with rhIL-7-hyFc treatment (day +24, median recipient T cells: 11,359 cells/μl, median mCART19: 964 cells/μl), but we did not see a rejection of the CAR T cells, and B cell aplasia in the mice was sustained throughout the duration of this experiment.

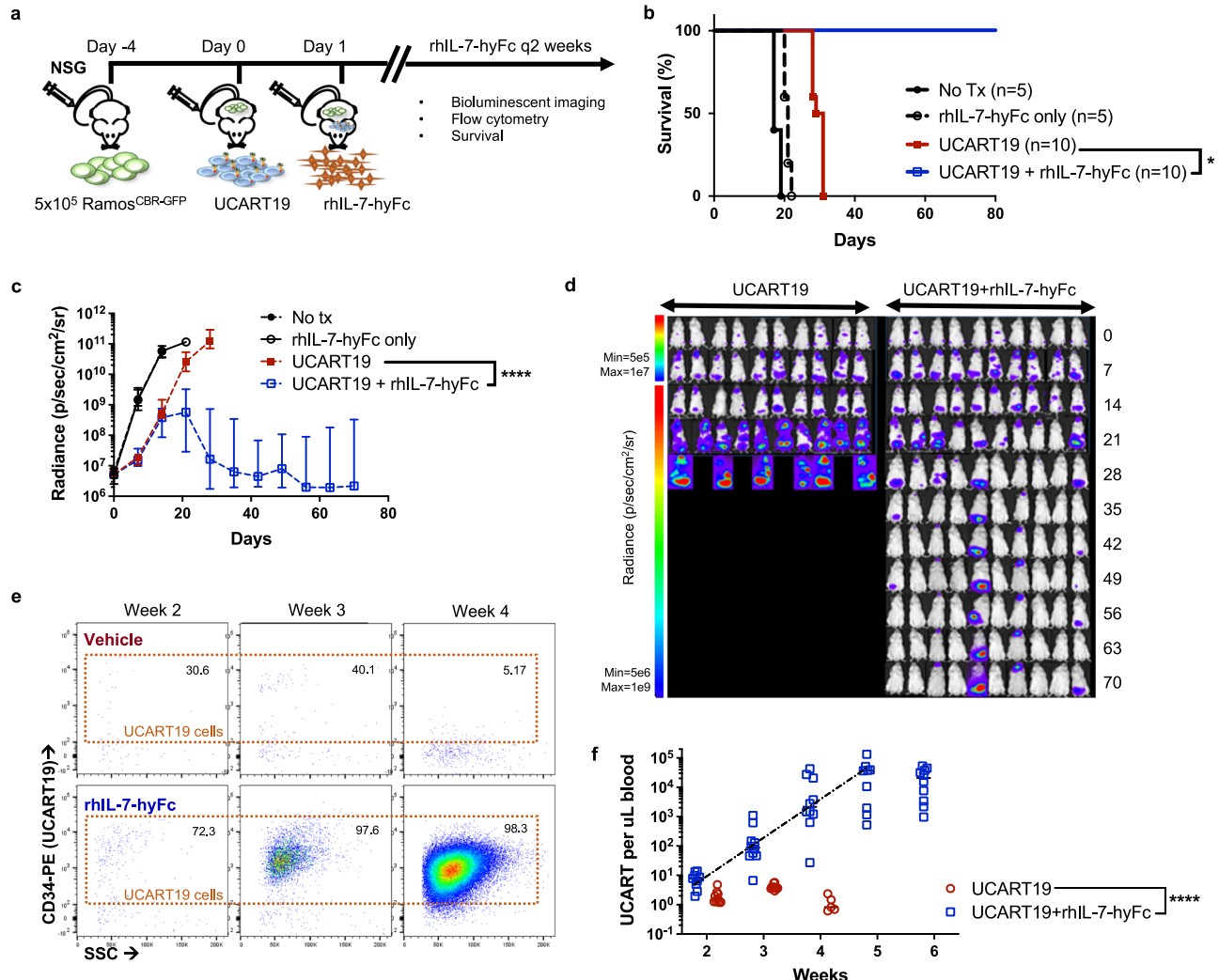

**Fig. 2 rhIL-7-hyFc dramatically enhances UCART19 expansion, persistence, and anti-tumor efficacy in NSG mice engrafted with CD19+ Ramos$^{CBR-GFP}$ cells in vivo. a** Experimental schema: NSG mice were injected via tail vein with $5 \times 10^5$ Ramos B lymphoma cells expressing click beetle red luciferase (CBR) and green fluorescent protein (GFP) (Ramos$^{CBR-GFP}$) on day −4, followed by $1 \times 10^6$ UCART19 on day 0, and rhIL-7-hyFc 10 mg/kg subcutaneously every 2 weeks for a total of three doses starting day +1 ($n = 5$ for no tx and rhIL-7-hyFc only groups; n = 10 for UCART19 and UCART19 + rhIL-7-hyFc groups). **b** Survival analysis for each treatment group. $p$ Values were calculated using two-sided Wilcoxon test (UCART19 + rhIL-7-hyFc vs. UCART19 alone, $p = 0.018$). **c** Tumor burden measured by bioluminescent imaging (BLI). Data represent median ± 95% CI. Two-tailed $p$ values were calculated using linear mixed model for repeated measurement data followed by post hoc multiple comparisons for between-group differences (UCART19 + rhIL-7-hyFc vs. UCART19 alone, $p < 0.0001$). **d** BLI images of individual mice treated with UCART19 only (left) and UCART19 + rhIL-7-hyFc (right). **e, f** Quantitative flow cytometric analyses of UCART19 expansion in peripheral blood. Representative FACS plots (**e**) and absolute UCART19/µl counts from peripheral blood (**f**). Each data point represents one mouse. Two-tailed p-values were calculated using a linear mixed model for repeated measurement data followed by post hoc multiple comparisons for between-group differences (UCART19 + rhIL-7-hyFc vs. UCART19 alone, $p < 0.0001$). Source data are provided as a Source data file. *$p \leq 0.05$, ****$p \leq 0.0001$.

**rhIL-7-hyFc improves survival of CD19+ tumor-bearing Balb/c mice after mCART19.** To confirm that rhIL-7-hyFc enhances CAR T cell anti-tumor activity in an immunocompetent host, we transitioned to using a B cell lymphoma cell line, A20, that expresses murine CD19, engrafted into Balb/c mice. To enable tracking of the tumor by BLI we used A20 cells transduced with CBR-GFP, which requires sublethal irradiation of the mice prior to prevent rejection of the tumor by endogenous immune cells. Mice received A20$^{CBR-GFP}$ cells either $5 \times 10^5$ intravenously (IV) or $1 \times 10^6$ subcutaneously (SC), and 7 days later were given $5 \times 10^5$ mCART19 cells without further conditioning, followed by rhIL-7-hyFc 10 mg/kg on days +1 and +15 (Fig. 5a). Due to prolonged thrombocytopenia in the mice after irradiation we were unable to evaluate peripheral blood CAR T cell expansion at

early time points; however, we did confirm on day +57 that mice that had received rhIL-7-hyFc had higher numbers of circulating CAR T cells as compared to mCART19 alone (Fig. 5b). In both the IV and SC tumor models, the tumor was better controlled with rhIL-7-hyFc, which led to improved survival in mice that received rhIL-7-hyFc after mCART19 treatment (Fig. 5c, d). Additional evaluation of T cell phenotype on day +57 by flow cytometry revealed that in contrast to the endogenous T cells, which were predominantly naive T cells, mCART19 cells were mostly comprised of memory T cells (Fig. 5e and Supplemental Figs. 8 and 9). The addition of rhIL-7-hyFc did not significantly change the proportion of T cell subsets for the most part, although there was a slight increase in effector CAR T cells and a decrease in central memory CAR T cells. In general, we observed

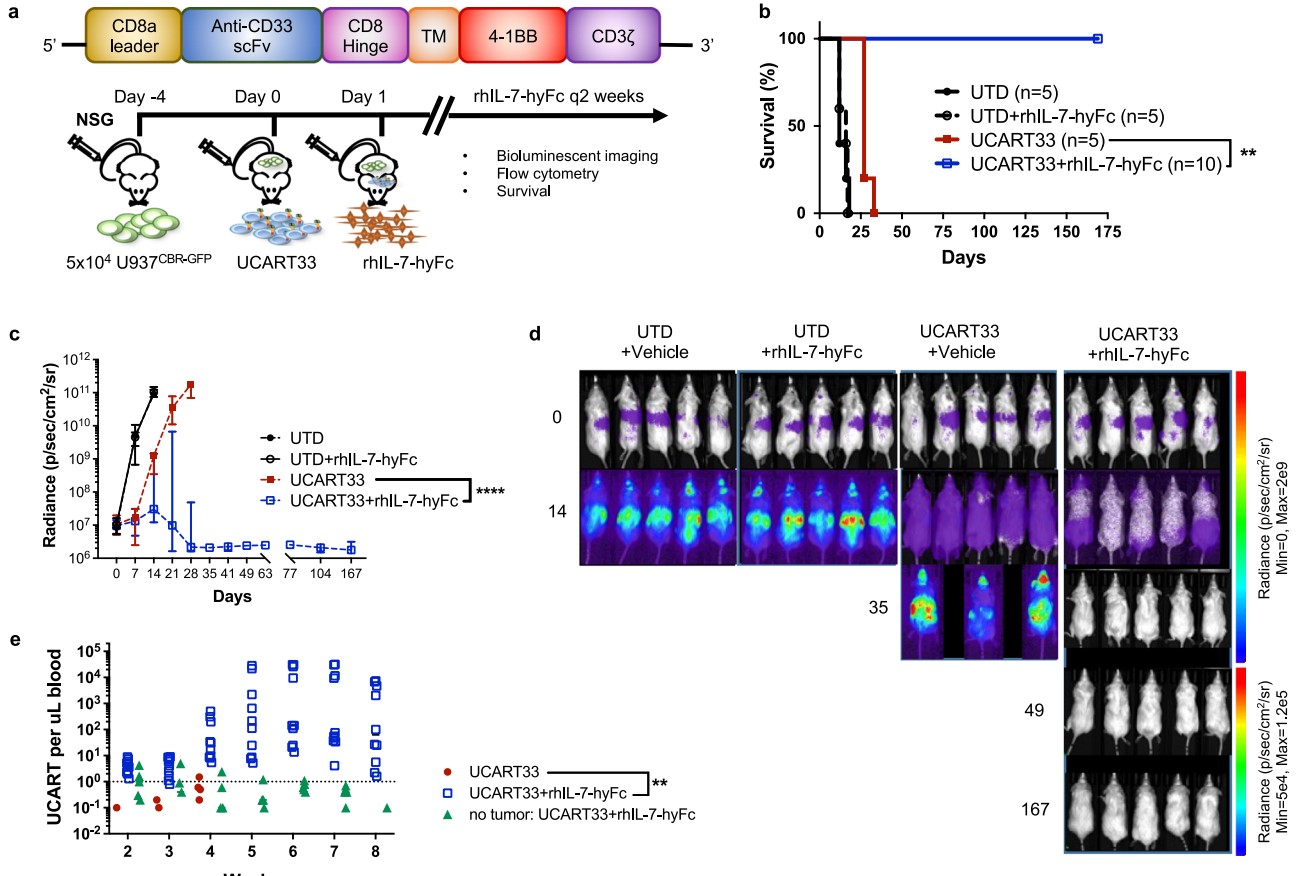

**Fig. 3 rhIL-7-hyFc enhances UCART33 expansion, persistence and anti-tumor efficacy in NSG mice engrafted with CD33+ U937^CBR-GFP cells in vivo.**
**a** UCART33 are human T cells gene-edited to delete *TRAC* and then transduced with a lentivirus carrying a second-generation CAR construct targeting CD33 with a 4-1BB endodomain. NSG mice were injected with $5 \times 10^4$ U937^CBR-GFP four days prior to administration of either $1 \times 10^6$ UTD (untransduced T cells) or UCART33 cells, followed by rhIL-7-hyFc 10 mg/kg subcutaneously on days +1, +15, and +29 ($n = 5$/group for UTD, UTD + rhIL-7-hyFc, UCART33, $n = 10$ for UCART33 + rhIL-7-hyFc). **b** Survival analysis for each treatment group. *p* Values were calculated using two-sided Wilcoxon test. (UCART33 + rhIL-7-hyFc vs. UCART33 alone, $p = 0.0015$). **c**, **d** Serial tumor measurements by BLI. Data represent median ± 95% CI. Two-tailed *p* values were calculated using a linear mixed model for repeated measurement data followed by post hoc multiple comparisons for between-group differences (UCART33 + rhIL-7-hyFc vs. UCART33 alone, $p < 0.0001$). **e** Quantitative flow cytometric analyses of UCART33 expansion in peripheral blood; also shown is a subgroup of mice ($n = 5$) that received UCART33 and rhIL-7-hyFc without prior tumor injection. Two-tailed *p* values were calculated using a linear mixed model for repeated measurement data followed by post hoc multiple comparisons for between-group differences (UCART33 + rhIL-7-hyFc vs. UCART33 alone, $p = 0.0053$). Source data are provided as a Source data file. **$p \leq 0.01$, ****$p \leq 0.0001$.

better expansion of CAR T cells as compared to endogenous T cells at this later time point, with the largest fold-change seen in the CD4+ effector and effector memory CAR T cells after rhIL-7-hyFc (Fig. 5f). These findings corroborate our findings in human CAR T cells, in that rhIL-7-hyFc leads to increased CAR T cell numbers, and this is more prominent within the effector and effector memory cell populations.

Long-term surviving mice after mCART19 or mCART19+rhIL-7-hyFc were protected from rechallenge with IV A20 tumor cells on day +100, regardless of whether the initial tumor was given IV or SC (Fig. 5g). Notably, on day +28 post-rechallenge (day +128 from initial T cell infusion), mice that had been previously treated with mCART19+rhIL-7-hyFc continued to have significantly higher numbers of CAR T cells detected in the peripheral blood than mice that had received mCART19 alone (Fig. 5h).

**rhIL-7-hyFc reduces the minimum effective dose of CART19.** Given the pronounced effect of rhIL-7-hyFc on CAR T cell expansion and efficacy, we hypothesized that rhIL-7-hyFc could reduce the minimum effective dose of CAR T cells required for

activity. To test this, we injected Balb/c mice with A20 tumor cells, followed by Cytoxan conditioning, and $5 \times 10^4$, $2.5 \times 10^5$, or $1.25 \times 10^6$ mCART19 cells, with or without rhIL-7-hyFc treatment (Fig. 6a). Due to the anti-tumor effects of the Cytoxan conditioning, the majority of the mice survived long-term in this experiment (Fig. 6b).

The addition of rhIL-7-hyFc increased the numbers of CAR T cells across all dose levels (Fig. 6c). Using the highest dose of $1.25 \times 10^6$ mCART19 cells as a benchmark, we found that rhIL-7-hyFc increased both the peak expansion and persistence of the next dose level of $2.5 \times 10^5$ cells beyond that seen with a five-fold higher dose of mCART19 alone. However, at the lowest dose of $5 \times 10^4$ mCART19 cells, the addition of rhIL-7-hyFc only transiently increased peak levels of mCART19. Using circulating B cells as a biomarker of CAR T cell activity, we found that without rhIL-7-hyFc, mice treated with $2.5 \times 10^5$ mCART19 recovered B cells after 3 weeks, while the addition of rhIL-7-hyFc led to prolonged B cell aplasia. Notably, control mice had a massive increase in B cell numbers after rhIL-7-hyFc, but mice treated with $2.5 \times 10^5$ or more mCART19 with rhIL-7-hyFc were still able to completely eliminate all detectable circulating B cells.

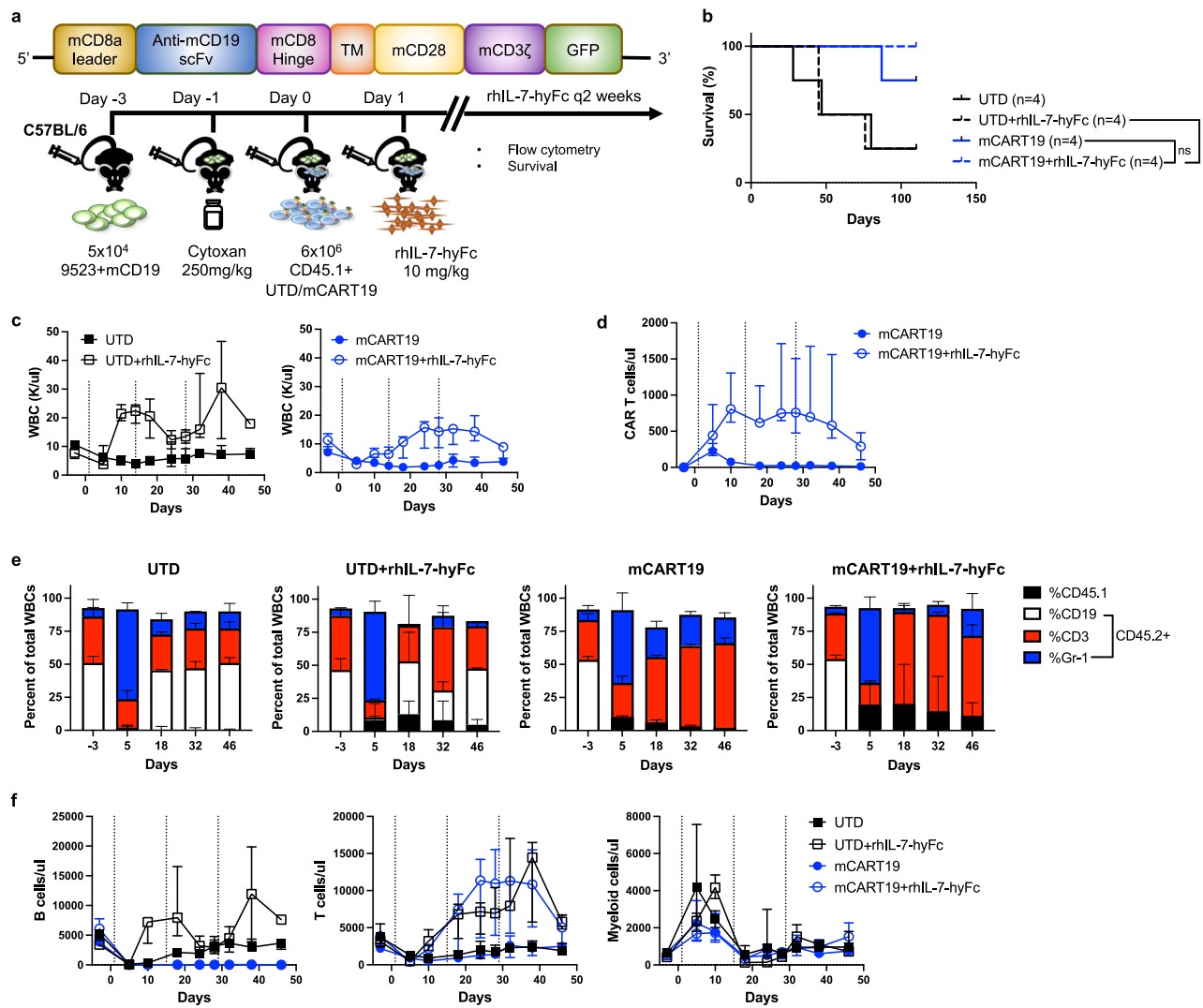

**Fig. 4 rhIL-7-hyFc enhances mCART19 expansion and persistence in immunocompetent mice. a** C57BL/6J mice were injected with $5 \times 10^4$ syngeneic tumor cells (9523, a myeloid cell line to which murine CD19 expression was added) followed by cyclophosphamide (Cytoxan) conditioning one day prior to receiving $6 \times 10^6$ UTD or murine CART19 (mCART19), then given serial injections of rhIL-7-hyFc on days +1, +15, and +29 ($n = 4$/group). **b** Overall survival of mice in each group. $p$ Values were calculated using two-sided Wilcoxon test (mCART19 + rhIL-7-hyFc vs. mCART19, $p = 0.32$, mCART19+rhIL-7-hyFc vs. UTD + rhIL-7-hyFc, $p = 0.044$). **c** Peripheral blood total WBC counts over time. **d** Absolute numbers of mCART19 cells over time. **e** Total WBC counts were fractionated by flow cytometry into percent CD45.1+ (adoptively transferred T cells) and CD45.2+ (endogenous cells); CD45.2 cells are subdivided into CD19+ B cells, CD3+ T cells, and Gr-1+ myeloid cells at the indicated timepoints. **f** Absolute numbers of CD45.2+ B cells, T cells, and myeloid cells over time, calculated by multiplying the total WBC count by the fraction of each cell determined by flow cytometry. All data represent median ± range. Dotted lines represent rhIL-7-hyFc injection time points. Source data are provided as a Source data file. ns: not significant, *$p \leq 0.05$.

Also, while the $5 \times 10^4$ mCART19 + rhIL-7-hyFc-treated mice were not able to clear B cells at any time point, these mice did not exhibit the marked increase in B cell numbers seen in the control group, suggesting that even this low cell dose had some effect against target cells when augmented by rhIL-7-hyFc (Fig. 6d). B cell aplasia was sustained beyond 100 days in mice receiving $2.5 \times 10^5$ mCART19 or higher with rhIL-7-hyFc, while mice that received any dose of mCART19 alone had B cell recovery (Fig. 6e). These results indicate that while a minimum dose of mCART19 is still required to achieve benefit from rhIL-7-hyFc, adding this reagent can lower the effective dose of mCART19 by at least fivefold, and enhance the durability of this treatment.

To ensure that these dose-sparing effects of rhIL-7-hyFc were also applied to human CAR T cells, we injected Ramos^CBR-GFP engrafted NSG mice with limiting doses of UCART19 starting at $5 \times 10^4$ cells followed by rhIL-7-hyFc treatment. Similar to the

murine CAR T cell studies, the $5 \times 10^4$ dose UCART19 did not have any effect on the tumor even with the addition of rhIL-7-hyFc. However, $2.5 \times 10^5$ UCART19 cells combined with rhIL-7-hyFc achieved tumor control in 3/5 mice, in contrast to mice given $2.5 \times 10^5$ UCART19 alone, where all mice had rapid tumor progression and died. As expected, the highest dose of $1.25 \times 10^6$ UCART19 in combination with rhIL-7-hyFc mediated tumor regression in all mice (Fig. 6f). Again, the degree of tumor control and survival seen with $2.5 \times 10^5$ UCART19 combined with rhIL-7-hyFc surpassed that seen with fivefold higher doses of UCART19 alone, indicating that rhIL-7-hyFc may allow for treatment with much lower CAR T cells doses than conventionally used, and still lead to better outcomes.

**In vivo treatment with rhIL-7-hyFc enhances both the quantity and quality of CAR T cells**. To determine whether the effects of

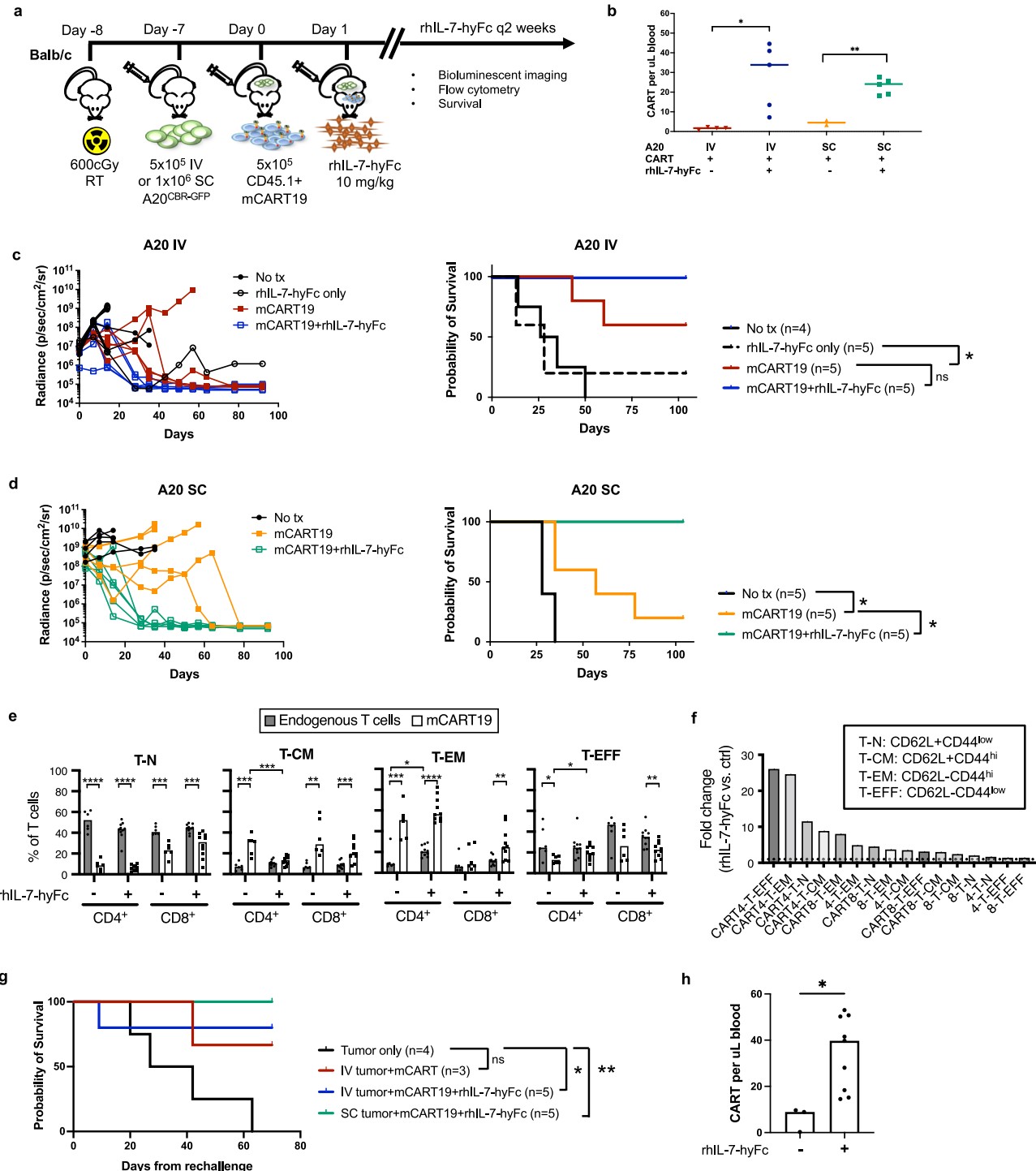

rhIL-7-hyFc were mediated solely by the quantitative increase in CAR T cell numbers, or whether there were also differences in the functionality CAR T cells, we harvested CAR T cells from mice after in vivo rhIL-7-hyFc treatment and reassessed their properties in vitro. Balb/c mice were given Cytoxan conditioning 1 day prior to receiving $2 \times 10^6$ UTD or mCART19 cells and injected with a single dose of rhIL-7-hyFc 10 mg/kg on day +1. Spleens were harvested from mice at 7 and 14 days after T cells. Spleen size and cell numbers were increased in both control and mCART-treated mice after rhIL-7-hyFc (Fig. 7a). Fractionation of cells within the spleen again revealed that expansion of the endogenous lymphoid

compartment drives the majority of responses to rhIL-7-hyFc in immunocompetent mice (Fig. 7b). Absolute numbers of mCART19 cells in the spleen were higher after rhIL-7-hyFc treatment, with average cell numbers increasing from day +7 to day +14, in contrast to vehicle-treated mice where mCART19 numbers decreased during this time (Fig. 7c). As expected, rhIL-7-hyFc decreased apoptosis (Fig. 7d) and increased proliferation (Fig. 7e) of the adoptively transferred T cells. Evaluation of cytotoxicity (Fig. 7f) and intracellular cytokine production (Fig. 7g) of purified mCART19 cells against A20 tumor cells revealed enhanced effector functions after exposure to rhIL-7-hyFc.

**Fig. 5 rhIL-7-hyFc improves the anti-tumor efficacy of mCART19 in immunocompetent mice. a** Balb/c mice were given sublethal irradiation (600 cGy), followed by injection with a syngeneic B cell lymphoma cell line, A20$^{CBR-GFP}$, either $5 \times 10^5$ cells intravenously (IV) or $1 \times 10^6$ cells subcutaneously (SC), then given $5 \times 10^5$ mCART19 cells and serial injections of rhIL-7-hyFc on days +1 and +15 ($n = 4-5$/group). **b** Absolute numbers of peripheral blood CAR T cells in each group measured at day +57. Lines represent median values, with each data point representing one mouse (IV A20 + CART + rhIL-7-hyFc vs. CART only, $p = 0.017$, SC A20 + CART + rhIL-7-hyFc vs. CART only, $p = 0.002$). **c** For mice given A20 IV, BLI measurements of tumor burden (left, each line represents one mouse), and overall survival (mCART19 vs. mCART19+rhIL-7-hyFc, $p = 0.13$, mCART19 vs. rhIL-7-hyFc only, $p = 0.04$). **d** For mice given A20 SC, BLI measurements of tumor burden (left, each line represents one mouse), and overall survival (mCART19 vs. mCART19+rhIL-7-hyFc, $p = 0.016$, mCART19 vs. no treatment, $p = 0.018$). **e, f** Day +57 peripheral blood flow cytometry evaluation of naive (T-N), central memory (T-CM), effector memory (T-EM), and effector (T-EFF) T cell populations within endogenous and CAR T cells with and without rhIL-7-hyFc, subdivided by CD4+ and CD8+ T cells. For this analysis, values from mice that received IV and SC tumor were pooled for each group (no rhIL-7-hyFc: $n = 6$, +rhIL-7-hyFc: $n = 10$). T cell memory subsets were defined using expression of CD62L and CD44 as indicated in **f**. **e** Proportions of T-N, T-CM, T-EM, and T-EFF within the CD4+ and CD8+ T cells in endogenous T cells and mCART19. Bar graphs represent median values, with each data point representing one mouse. *p* Values are provided in the Source data file. **f** Fold-change of T cell subsets were calculated by averaging the absolute numbers of each cell subset, and dividing the numbers for mCART19+rhIL-7-hyFc treated mice over mCART19 only controls. CART4 = CD4+ CAR T cells, CART8 = CD8+ CAR T cells, 4 = endogenous CD4+ T cells, 8 = endogenous CD8+ T cells. **g** Long-term surviving mice were rechallenged with $1 \times 10^6$ A20 cells (without CBR-GFP) IV on day +100. A separate cohort of mice with no prior treatment was also injected with A20 tumor cells on the same day as controls ($n = 4$). **h** Peripheral blood CAR T cell numbers at day +28 post-rechallenge. mCART19+rhIL-7-hyFc ($n = 9$) vs. mCART19 ($n = 3$), $p = 0.014$. *p* values for survival curves were calculated using two-sided log-rank test, and for **b, e, h** by two-sided unpaired Student's *t* test. Source data are provided as a Source data file. No tx: no treatment, ns: not significant, $*p \leq 0.05$, $**p \leq 0.01$, $***p \leq 0.001$, $****p \leq 0.0001$.

We also evaluated the cytotoxicity of UCART19 cells extracted from the spleens of NSG mice treated with rhIL-7-hyFc. Mice that received rhIL-7-hyFc had markedly enlarged spleens as compared to control mice (Fig. 7h), with UCART19 comprising >90% of splenocytes (data not shown). As spleens from control mice had insufficient UCART19 cells to assay, the efficacy of UCART19 killing after in vivo rhIL-7-hyFc treatment was compared to cryopreserved cells from the same batch of UCART19 infused into the mice on day 0 (input). In vivo rhIL-7-hyFc treated UCART19 displayed significantly greater cytotoxicity than input UCART19 against CD19+ Ramos and Nalm6 tumor cell lines (Fig. 7i, j), with input controls requiring 2–4-fold greater numbers of effector cells to achieve equivalent cytotoxicity to UCART19 from rhIL-7-hyFc treated mice. These results indicate that rhIL-7-hyFc not only increases CAR T cell numbers but also enhances the effector functions of CAR T cells against the tumor.

**Single-cell RNA sequencing (scRNA-seq) demonstrates that rhIL-7-hyFc preferentially expands IL7R+ effector memory CAR T cells.** To further elucidate the effects of rhIL-7-hyFc on CAR T cell phenotype, scRNA-seq was performed on UCART19 derived from spleens of rhIL-7-hyFc treated NSG mice (Fig. 8a). Due to low cell numbers and poor viability, we were unable to analyze UCART19 cells collected from Ramos engrafted NSG mice given UCART19 alone, so cryopreserved cells from the same batch of UCART19 infused into the mice on day 0 were used as the comparator group to rhIL-7-hyFc treated UCART19. After stringent filtering, a total of 27,676 cells were identified, with 2489 cells derived from day +14 (week 2), 16,367 cells from day +28 (week 4), and 7911 cells from the UCART19 input cells (Input). The integration of all cells using Seurat identified a total of 15 unique T cell clusters, with varying proportions of cells mapped back to each treatment time point (Fig. 8b, c). While most T cell clusters were maintained across treatment timepoints, clusters 0 and 1 showed a dramatic increase at week 4. Downsampled UMAP projections ($n = 2000$ cells) further highlight the predominance of clusters 0 and 1 at week 4, relative to input and week 2 (Fig. 8d).

Previous studies have established the alternative splicing repertoire of naive and memory T cells involving exons 4, 5, and 6 of PTPRC, where isoforms RBC and RAB are specific to naive T cells, the RB isoform is shared by naive and memory T cells, and RO is specific to memory T cells[11,12]. To evaluate the T cell representation within our samples, we quantified the

isoform expression of *PTPRC* (CD45), by utilizing sjcount (https://github.com/pervouchine/sjcount) to quantify splice junction counts across the scRNA-seq dataset (Supplemental Fig. 10). By comparing the proportion of overall reads that spanned exon 3 and downstream exons 4, 5, or 7, the majority of reads could be identified as either RO or RB/RBC. To evaluate whether the predominant isoform was RB or RBC, we then quantified the total reads mapped from exon 5 to downstream exons 6 or 7, and found a majority of reads supported the RB isoform. Therefore, isoform analysis of *PTPRC* provides additional evidence that the vast majority of UCART19 after rhIL-7-hyFc treatment are memory T cells.

We then mapped the *PTPRC* isoforms back to the T cell clusters identified by Seurat, and found that cluster 4 had a high fraction of the RBC isoform specific to naive T cells. After determining the likely naive T cell group, we then performed trajectory analysis using Monocle3 to evaluate the developmental transitions between T cell clusters. A UMAP of all T cell clusters and the overlapping trajectory indicates that cells starting with cluster 4 (naive T cells) transition to several downstream clusters, including cluster 1 and cluster 0, which harbor more of an effector memory state, before transitioning to a number of additional T cell states (Fig. 8c).

To further characterize the different T cell clusters, we annotated each cluster by the expression of well-known T cell markers: *CD4, CD8A, CCR7*, and *SELL*[13]. Most of the CD8 clusters had an effector memory phenotype (clusters 1,2,3,8; *CD8A+/CCR7−/SELL−*), while one cluster had a naive/central memory phenotype (cluster 4; *CD8A+/CCR7+/SELL−*), and one was intermediate (cluster 10; *CD8A+/CCR7−/SELL+*). Cluster 10 also had high expression of *CD27* and *PTPRC* isoform RB (CD45RB), which has been reported in long-lived memory T cell groups[11]. Similarly, for the CD4 clusters, the majority of cells were effector memory (clusters 0,6,9; *CD4+/CCR7−/SELL−*), while one cluster was distinctly central memory (cluster 7; *CD4+/CCR7+/SELL+*). All of the remaining populations (clusters 5, 11, 12, 13, 14) lacked expression of *CCR7* and *SELL* (Fig. 8f).

We also calculated the exhaustion scores[14–16] of the T cells using a Seurat scoring function based on the expression of the following well-known exhaustion markers: *CTLA4, PDCD1, LAG3, HAVCR2, CD160, CD244*, and *TIGIT*. The overall exhaustion score was strikingly higher in the input sample and decreased with successive treatment timepoints, leaving only a subset of cells in cluster 10 at week 4 containing an exhaustive T

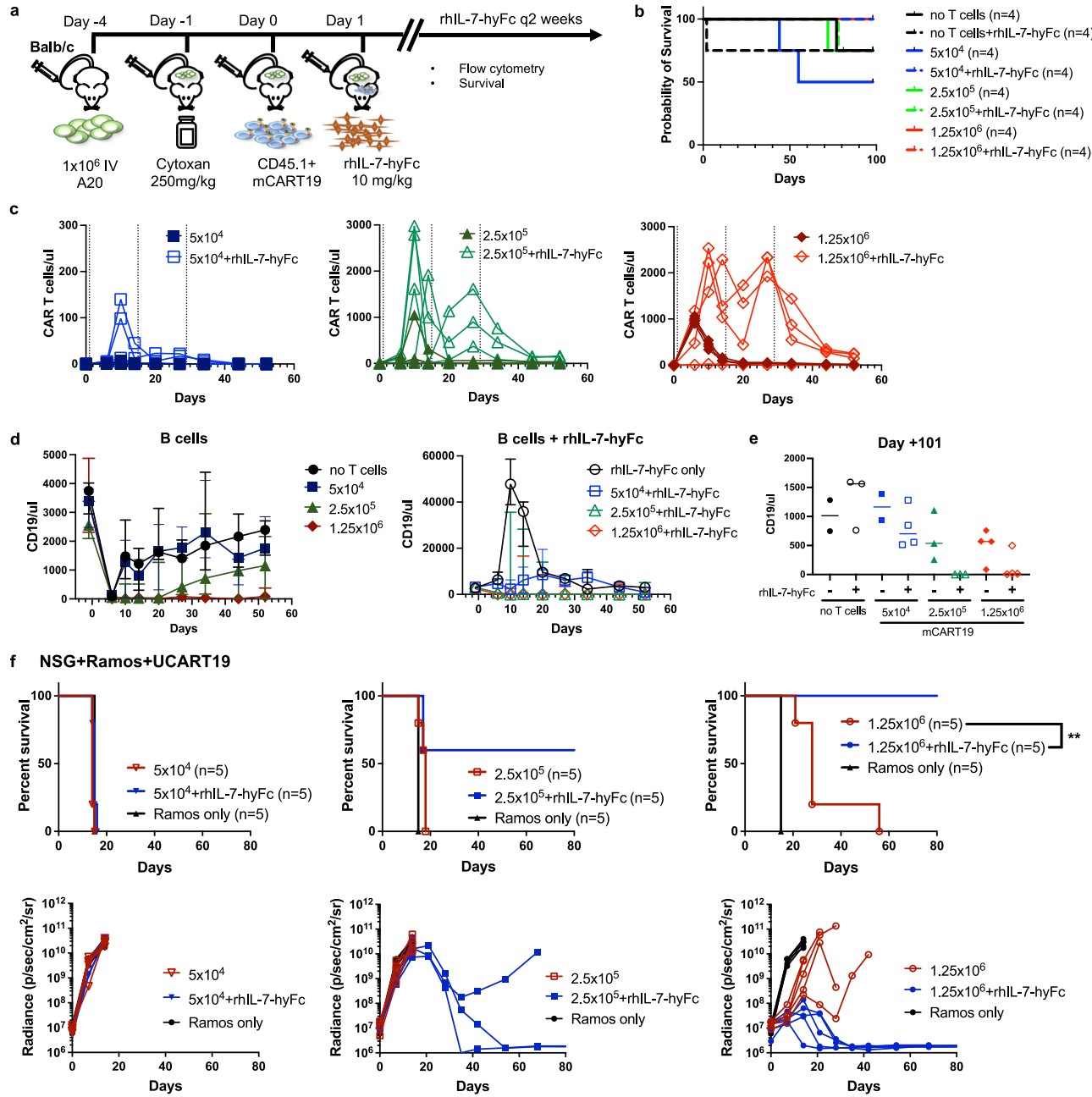

**Fig. 6 rhIL-7-hyFc reduces the number of CAR T cells required for the biological effect. a–e** Balb/c mice were first given $1 \times 10^6$ A20 tumor cells, followed by cyclophosphamide (Cytoxan) conditioning 1 day prior to receiving mCART19 ($5 \times 10^4$, $2.5 \times 10^5$, or $1.25 \times 10^6$ cells per mouse), then given serial injections of rhIL-7-hyFc on days +1, +15, and +29 ($n = 4$/group). **b** Overall survival of mice for all groups. **c** Peripheral blood CAR T cell expansion was measured over time by quantification of CD45.1+GFP+ cells. Each line represents values from one mouse. **d** B cell numbers were measured over time by quantification CD45.2+ CD19+ cells. Data represented as median ± range. Note that $5 \times 10^4$ cells (left panel) are depicted on a different scale. **e** B cell numbers on day +101 after T cell injection. **f** NSG mice were injected with $5 \times 10^5$ Ramos[CBR-GFP] followed by UCART19 treatment ($5 \times 10^4$, $2.5 \times 10^5$, or $1.25 \times 10^6$ cells per mouse) with serial injections of rhIL-7-hyFc on days +1, +15, and +29 ($n = 5$ mice per group). The top panels represent overall survival, while the bottom panels show serial tumor burden measurements (each line represents one mouse). $p$ Values were calculated using two-sided Wilcoxon test (UCART19 $1.25 \times 10^6$ + rhIL-7-hyFc vs. UCART19 $1.25 \times 10^6$ only, $p = 0.0039$). Source data are provided as a Source data file. **$^{**}p \leq 0.01$.**

cell phenotype (Fig. 8e). Of note, cluster 10 exhibited a high exhaustion score but also maintained expression of *GZMK*, *TIGIT* and *PDCD1*, supportive of an exhausted-like memory state[17] (Fig. 8e, f). Additionally, in clusters 0 and 1 that were more highly represented at week 4, increased expression of *IL7R*, *GZMB*, *S100A4* and persistent high expression of *PTPRC* was observed, providing further evidence that effector memory phenotypes are maintained by in vivo administration of rhIL-7-hyFc (Fig. 8g).

## Discussion

Enhancement of CAR T cell activity has the potential to improve patient outcomes by reducing target positive relapse in patients with CD19+ B cell malignancies. We show that adding rhIL-7-hyFc to CAR T cells improves T cell expansion, persistence, and anti-tumor activity, both with human CAR T cells targeting CD19 or CD33 in an NSG xenograft model and with murine CAR T cells targeting CD19 in two different strains of immunocompetent mice.

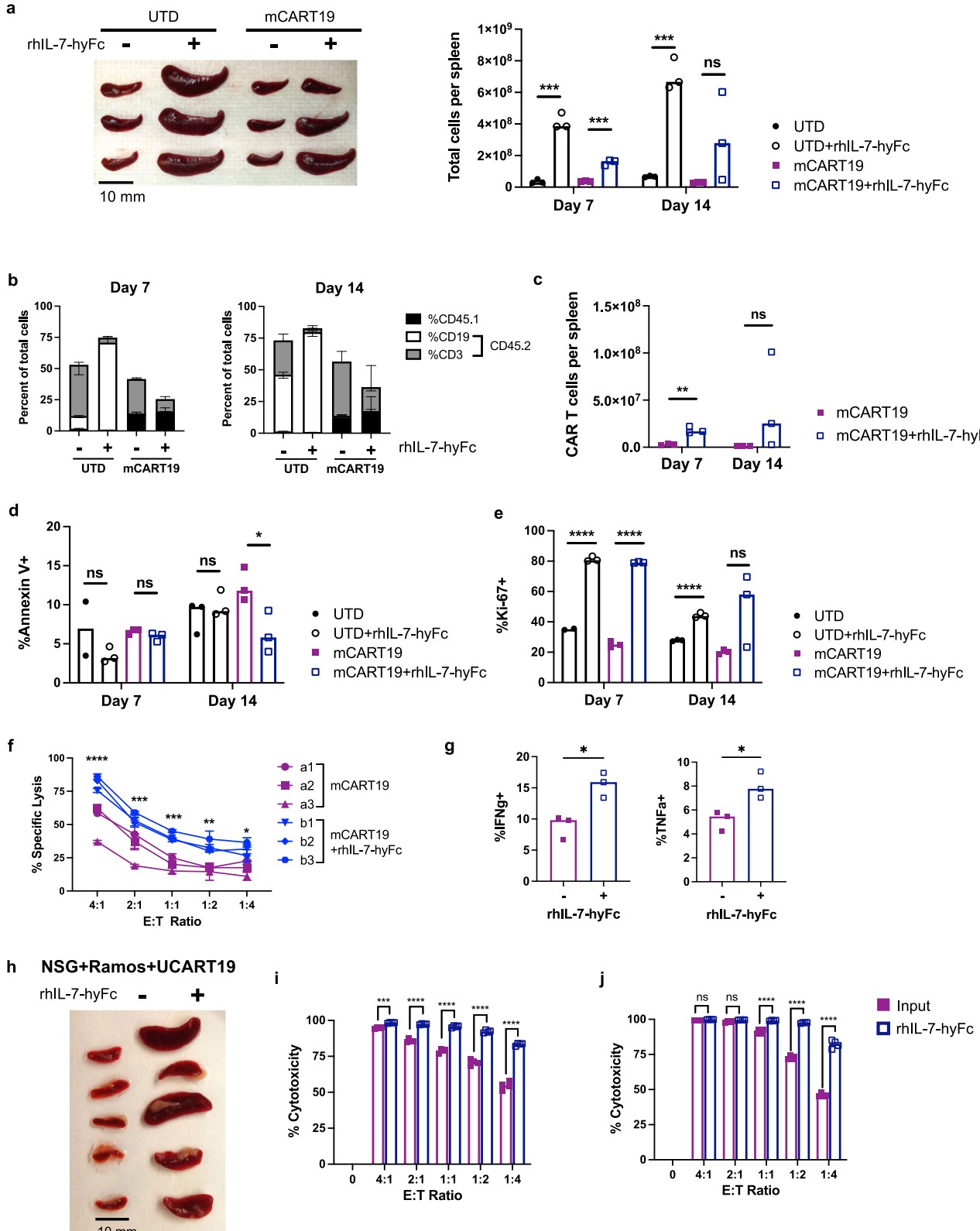

IL-7 is well known to promote T cell proliferation and expand the T cell repertoire. Clinical trials exploring the use of rhIL-7 for a variety of indications have shown that it is well tolerated and increases T cell numbers in human subjects[18–22]. However, one of the factors limiting the development of rhIL-7 has been its short half-life in vivo, necessitating frequent doses to maintain biological activity. We used a novel form of IL-7 that is fused to a hybrid Fc domain, extending its half-life in vivo[23]. In vitro studies comparing rhIL-7-hyFc and rhIL-7 show that they have nearly identical effects on CAR T cells, as both increases the potency and duration of CAR T cell activity. While we have not performed in vivo comparison studies, it is likely that rhIL-7 would also mediate similar effects to rhIL-7-hyFc in this setting if given frequently enough, but the convenience of the long-acting

**Fig. 7 In vivo rhIL-7-hyFc therapy enhances both the quantity and quality of CAR T cells. a–g** Balb/c mice were given cyclophosphamide 250 mg/kg on day −1, then $2 \times 10^6$ CD45.1+ UTD or mCART19 cells on day 0, followed by rhIL-7-hyFc on day +1. Spleens were harvested on days +7 and +14 after T cell injection ($n = 3$/group). Bar graphs represent median values, with each data point representing one mouse. **a** Spleen sizes (left, day +14) and cell numbers (right) were significantly increased in mice given rhIL-7-hyFc. **b** Fractionation of spleen cell subsets shows that the majority of cells are derived from the CD45.2+ recipient. **c** Absolute numbers of mCART19 cells were higher in the spleens of mice after rhIL-7-hyFc treatment. **d** Apoptosis was reduced in mCART19 at day +14 with rhIL-7-hyFc. **e** Both UTD and mCART19 show marked proliferation in response to rhIL-7-hyFc on day +7, which is better sustained at day +14 for mCART19. **f**, **g** On day +7 mCART19 was purified from spleens and subject to in vitro functional assays. **f** mCART19 cytotoxicity against A20 tumor cells was measured at various E:T ratios by BLI after 48 h of incubation. Each line represents mCART19 cells extracted from one mouse. **g** Intracellular IFNγ and TNFα production of mCART19 was measured after 4 h of incubation with A20 tumor cells. **h-j** NSG mice received $5 \times 10^5$ Ramos$^{CBR-GFP}$ followed by UCART19 treatment and rhIL-7-hyFc on days +1 and +15. Spleens were harvested from mice on day +28. **h** Spleens harvested from rhIL-7-hyFc-treated mice were notably larger than controls. **i**, **j** Target cell viability was determined by FACS-based cytotoxicity assay of UCART19 from day 0 (input) as compared to UCART19 derived from spleens of rhIL-7-hyFc treated mice. The original Ramos tumor cells (**i**) or an independent CD19+ B cell line, Nalm6 cells (**j**), were incubated with CD34-purified UCART19 at E:T ratios ranging from 4:1 to 1:4 for 20 h. Bar graphs represent median values, with each data point representing a technical replicate. All $p$ values were determined by two-sided unpaired Student's $t$ test. $p$ Values and source data are provided in the Source data file. ns: not significant, *$p \leq 0.05$, **$p \leq 0.01$, ***$p \leq 0.001$, ****$p \leq 0.0001$.

formulation would make this the preferred agent when moving into clinical studies in humans. On this note, we used a dose of 10 mg/kg rhIL-7-hyFc, to achieve maximum effect in these pre-clinical models. While it is likely that lower doses of rhIL-7-hyFc would be equally effective, the 10 mg/kg dose in mice is equivalent to 813 ug/kg in humans, and current clinical trials with rhIL-7-hyFc have used doses up to 1200 ug/kg, without any serious adverse events[24–26]. Therefore, our preclinical dosing can potentially be directly converted to equivalent dosing in human subjects.

In vivo studies with both human and mouse CAR T cells demonstrate that the addition of rhIL-7-hyFc can lead to a more than fivefold reduction of CAR T cell numbers required for biologic activity, with improved anti-tumor function and longer persistence. These findings are significant as currently CAR T cells are an autologous therapy, and the potency of each patient's CAR T cells can be quite variable[6,27]. While some patients' CAR T cells are effective against tumors by themselves, the majority of patients have less robust T cell function, and the addition of rhIL-7-hyFc may be able to achieve tumor regression in these individuals who would otherwise have succumbed to refractory disease. Additionally, if allogeneic CAR T cell products advance to the clinic in the future, rhIL-7-hyFc can be used to lower the minimum required dose of CAR T cells per patient, which could be a cost-effective strategy to overcome the financial toxicity of this therapy.

Given the potency of rhIL-7-hyFc to enhance CAR T cell numbers, we elected to use gene-edited TCR deficient CAR T cells to eliminate the confounding effects of xenogeneic GVHD in our preclinical mouse xenograft models. In terms of clinical translation, either autologous CAR T cells or allogeneic TCR deficient CAR T cells are used for patient care, so GVHD is not a concern. However, NSG mouse models fail to address the potential for host versus graft effect, specifically the concern that rhIL-7-hyFc may contribute to host rejection of CAR T cells, which can occur even in the autologous setting by immune responses directed against the CAR[28]. Therefore, we also tested rhIL-7-hyFc in immuno-competent mouse models, using congenic CD45.1+ mCART19 containing GFP as a marker gene. While rhIL-7-hyFc induced a preeminent expansion of endogenous lymphocytes, this did not lead to rejection of CAR T cells, as mCART19 persisted for many weeks and also mediated prolonged B cell aplasia up to 100 days after initial T cell injection. While the T cells themselves would not be foreign to the host, the presence of the synthetic CAR construct and GFP could theoretically lead to rejection, and so the persistence of CAR T cells in this setting was reassuring in that regard.

Importantly, we did not see any evidence of toxicity with the combination of mCART19 and rhIL-7-hyFc in the immuno-competent mice. Additionally, murine CAR T cell numbers in immunocompetent mice did not reach the astronomical levels seen with human CAR T cells in NSG mice after rhIL-7-hyFc. The NSG model provides proof of principle that this reagent is effective on human CAR T cells in vivo, but the lack of endo-genous lymphocytes in these mice is likely responsible for the massive expansion of CAR T cells, and in human subjects, the kinetics of CAR T cell expansion is expected to be more akin what was seen in the immunocompetent mice.

The ability of IL-7 to promote T cell memory is attractive to those working in the CAR T cell field. Culture of CAR T cells with IL-7 and IL-15 during manufacturing has been shown to enhance T cell differentiation toward a central memory (Tcm) phenotype, enhancing the persistence and anti-tumor efficacy of CAR T cells in vivo and prolonging survival in murine models of B-ALL[29]. We postulated that rhIL-7-hyFc would mediate enhanced CAR T cell function and anti-tumor activity through the preferential expansion and enhanced persistence of Tcm populations. How-ever, contrary to our expectations, both human and murine CAR T cells displayed a predominantly effector memory (Tem) phe-notype after in vivo rhIL-7-hyFc treatment. In the murine model, we were able to compare the response of endogenous T cells and CAR T cells to rhIL-7-hyFc. For endogenous T cells, the effects of rhIL-7-hyFc were approximately equivalent between CD4+ and CD8+ T cells, and led to preferential increase of memory T cells. However, in CAR T cells the effects of rhIL-7-hyFc were most prominent in the CD4+ effector cells. These differences may be due to the inherent bias of CAR T cells towards effector cells that are amplified by the presence of antigen-positive target cells providing additional activation and proliferation signals in vivo. Functional assessment of CAR T cells after rhIL-7-hyFc in vivo demonstrated better protection against tumor rechallenge and prolonged B cell aplasia as compared to CAR T cells alone, indicating that rhIL-7-hyFc does enhance immune protection, despite the absence of phenotypic Tcm generation.

Transcriptional analysis of UCART19 exposed to rhIL-7-hyFc in vivo by scRNAseq revealed overrepresentation of both CD4+ and CD8+ effector memory T cell subsets (clusters 0 and 1), as characterized by high expression of S100A4 and low expression of CCR7 and CD62L. By week 4, clusters 0 and 1 represented the majority of UCART19, with the expression of granzyme B and IL-7Rα increasing over time within these clusters. Additionally, T cell exhaustion status, as measured by expression of inhibitory receptors, decreased over time when UCART19 was exposed to rhIL-7-hyFc treatment. Based on this data, it is conceivable that

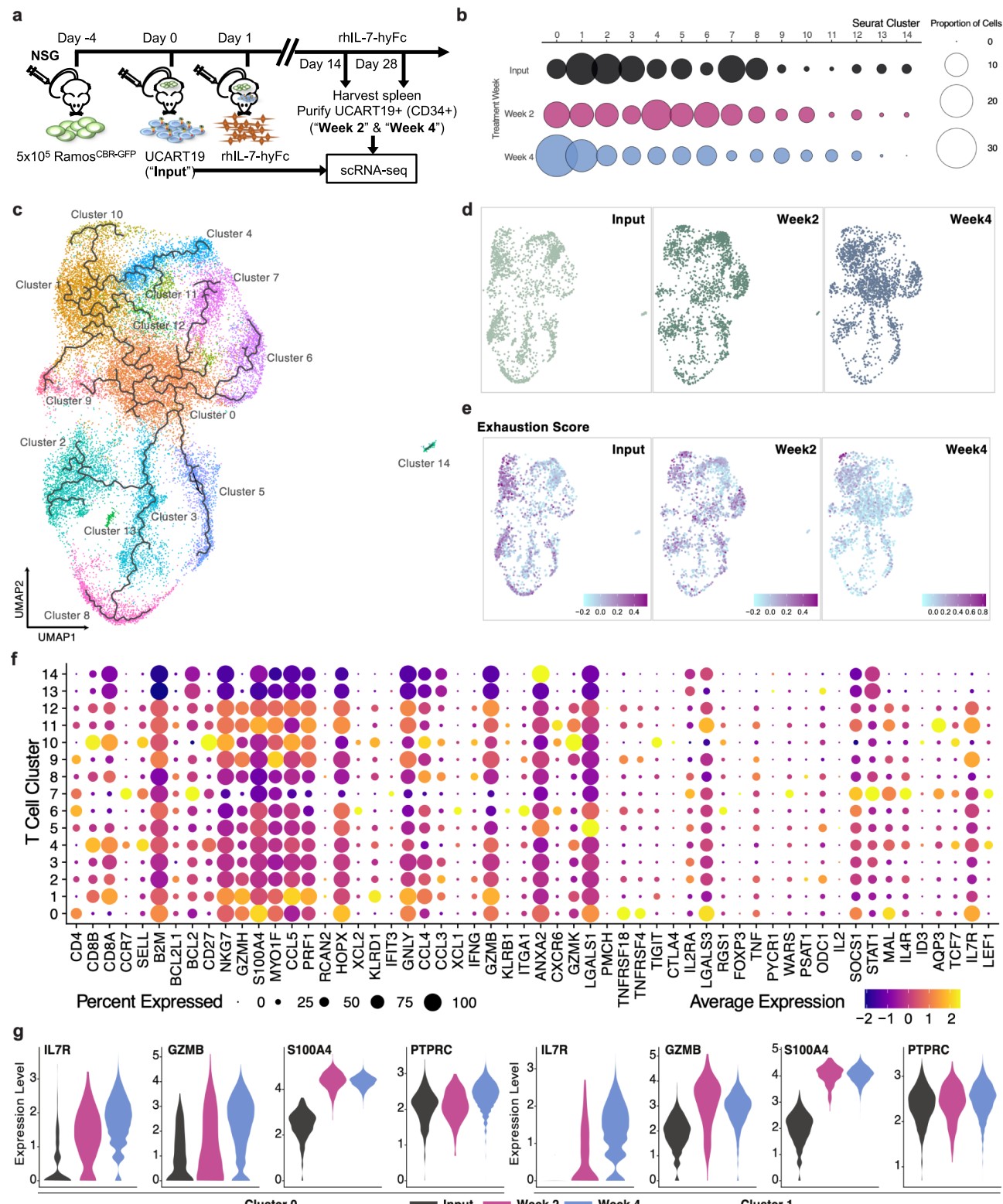

rhIL-7-hyFc therapy may prevent CAR T cell dysfunction and propagate a highly functional CAR T phenotype in vivo.

Other groups have adopted different strategies to enhance CAR T cell function using the IL-7 signaling pathway. These approaches range from engineering CAR T cells to secrete IL-7[30,31], adding inverted cytokine receptors in which the external domain from IL-4α is fused to the internal signaling domain from IL-7Rα[32], and expressing a constitutively active IL-7Rα[33]. However,

genetically engineering IL-7 signaling into T cells may lead to uncontrolled T cell expansion and proliferation, which may exacerbate CAR T cell toxicities such as cytokine release syndrome (CRS) and neurotoxicity, or even result in malignant transformation of the T cells themselves. Using rhIL-7-hyFc with CAR T cells provides opportunities to modify both the timing and dose of the reagent to mitigate toxicities while still enhancing therapeutic efficacy. Furthermore, rhIL-7-hyFc-mediated

**Fig. 8 Single-cell RNA sequencing reveals that rhIL-7-hyFc preferentially expands IL-7Rα effector memory CAR T cells. a** Ramos$^{CBR/GFP}$ bearing NSG mice ($5 \times 10^5$ cells IV on day −4) received rhIL-7-hyFc on days +1 and +15 following $1 \times 10^6$ UCART19 injection. Spleens were harvested on days +14 and +28, and UCART19 cells enriched using CD34+ magnetic selection were subject to single-cell RNA-sequencing. **b** Proportion of cells (size of circle) for each single-cell cluster (Seurat) were broken down by treatment group (Input, Week 2, and Week 4). **c** UMAP representation of the single-cell RNA-sequencing data, combined from all samples, colored and labeled by Seurat clusters. **d** Downsampled objects ($n = 2000$ cells for each treatment group) separated by treatment week. **e** Downsampled objects ($n = 2000$ cells for each treatment group) colored by exhaustion score, calculated based on the expression of genes known to be involved in T cell exhaustion (*CTLA4, PDCD1, LAG3, HAVCR2, CD160, CD244,* and *TIGIT*). Dark purple indicates a higher exhaustion phenotype, and light blue indicates a less exhaustive phenotype. **f** Gene expression analysis of relevant T cell genes and pathways for each Seurat cluster. The size of the circle indicates the percentage of cells with the expression of each gene and color indicates average expression across the cells in that unique cluster. **g** Focused view of clusters 0 and 1 showing increased expression of the genes IL-7Rα, GZMB, S100A4, and PTPRC across treatment groups.

enhancement of endogenous recipient T and NK cell proliferation and function may provide additional anti-tumor benefit.

In summary, we demonstrate that rhIL-7-hyFc can dramatically enhance CAR T cell expansion, persistence, and anti-tumor activity in vivo, resulting in significant improvement in survival in mice. These results are being pursued further in an ongoing clinical study (NCT05075603) to test the impact of rhIL-7-hyFc on CD19-targeting CAR T cells in humans.

## Methods

**rhIL-7-hyFc.** rhIL-7-hyFc was generously provided by NeoImmuneTech, Inc.

**CAR constructs.** The CAR19 construct was synthesized using the immunoglobulin heavy and light chain sequences of anti-human CD19 antibody clone FMC63, and cloned into PLVM lentiviral vector containing a third-generation CD28-4-1BB-CD3ζ CAR construct. Additionally, the vector was modified to express human CD34 via a P2A peptide. Expression of CD34 allowed for detection of the CAR following viral transduction, as well as purification of CAR positive cells using anti-human CD34 positive magnetic beads, as previously described[34]. The CAR33-4-1BB-CD3zeta construct was synthesized based on a previously published sequence[35] and also cloned into the PLVM lentiviral vector.

The mouse CAR19 vector was kindly provided by Dr. M. Sadelain (Memorial Sloan-Kettering Cancer Center, New York, NY). The CAR sequence is based on the immunoglobulin heavy and light chain sequences of anti-mouse CD19 antibody clone 1D3, followed by the mouse CD8 transmembrane region, mouse CD28 signal transduction domain, and the mouse CD3 cytoplasmic domain. Additionally, to facilitate the tracking of CAR transduced cells, GFP was linked to the CAR construct by a glycine-serine linker. This construct was cloned into the SFG retroviral vector[10]. All plasmids were verified by sequencing across the CAR construct prior to virus production.

**Viral vector production.** Lentivirus for human T cell transduction was generated by transfecting the Lenti-X 293T cell line (Clontech, 632180) with the PLVM CAR vector and packaging plasmids, pMD.Lg/pRRE, pMD.G, and pRSV.Rev using Lipofectamine 2000 (ThermoFisher Scientific, 11668500). The virus was harvested 24 and 48 h post transfection, filtered to remove cell debris, concentrated by ultracentrifugation for 90 min at 25,000 r.p.m. at 4 °C, and stored at −80 °C in single use aliquots.

Retrovirus for murine T cell transduction was generated by transfecting the Lenti-X 293T cells with the SFG CAR vector and the Ecopac packaging plasmid using Lipofectamine 2000. The virus was harvested 24 and 48 h post transfection, filtered to remove cell debris, stored at −80 °C, and thawed immediately prior to use. Both human and murine CAR T cell activities were verified by in vitro cytotoxicity assays against target-positive and target-negative cells prior to in vivo experiments.

**Human CAR T cells.** TRAC gRNA was commercially synthesized (Trilink Biotechnologies San Diego, CA), incorporating 2′-O-methyl and 3′ phosphorothioate (ps) bases at the three terminal bases of the 5′ and 3′ ends of the gRNA to protect from nuclease activity[36]. Streptococcus pyogenes Cas9 (spCas9) mRNA (5meC, Ψ) was also purchased from Trilink Biotechnologies. The full RNA guide sequence is as follows, with target sequence underlined: 5′_2′OMe(G(ps)A(ps)G(ps))AAUCA AAAUCGGUGAAUGUUUUAGAGCUAGAAAUAGCAAGUUAAAAUAAGGC UAGUCCGUUAUCAACUUGAAAAAGUGGCACCGAGUCGGUGC2′OMe (U(ps)U(ps)U(ps) U 3′.

T cells obtained from healthy human donors and were cultured in RPMI complete media (RPMI with 10% FBS, 1% PenStep, 1% Glutamax, 2 mM HEPES and 50 nM 2-Mecaptoethnol), supplemented with 50 U/mL IL-2, 10 ng/mL IL-15 and 10 ng/mL IL-7. CD3/CD28 Dynabeads (ThermoFisher Scientific, 40203D) were used to activate T cells, with a bead to cell ratio of 3:1. On day +2 post-activation,

beads were removed and $1 \times 10^7$ T cells were electroporated in 100 µl of MaxCyte EP buffer using 15 µg spCas9 mRNA and 20 µg of TRAC gRNA using the Maxcyte GT, program "Expanded T cell #2". Cells were transduced with lentiviral particles in the presence of 6 µg/ml polybrene (Sigma Aldrich. St Louis MO) at 1–24 h post electroporation. Cells were expanded for an additional 8 days, then depleted of residual non-edited TCR+ T cells using CD3+ magnetic bead-based on the AutoMacs (Miltenyi) per manufacturer protocol. Cells were either used fresh or frozen and thawed as needed for the experiments.

**Mouse CAR T cells.** Spleens were harvested from CD45.1+ mice and enriched for T cells using the Miltenyi Pan T cell isolation kit II (#130-095-130) per manufacturer protocol. Purified T cells were activated on the same day of harvest (=day 0) with CD3/CD28 Dynabeads (Gibco, 11453D) at 1:1 bead to cell ratio, and cultured in RPMI complete media supplemented with 10 U/ml IL-2 and 10 ng/ml IL-15. On days +1 and +2 post-activation, T cells were spinoculated with retroviral supernatant on plates coated with Retronectin (Clontech, T100B). On day +3, beads were removed and %GFP was measured by flow cytometry to assess the efficiency of gene transfer. Cells were either used fresh or frozen and thawed as needed for the experiments.

**Cell lines.** Ramos (CRL-1596), NALM6 (CRL-3273), U937 (CRL-1593.2), and A20 (TIB-208) cell lines were obtained directly from ATCC. The cell lines were mycoplasma tested and antigen expression was confirmed by flow cytometry prior to use. PLVM EF1α$^{CBR-GFP}$ lentivirus was used to transduce cells with the CBR-GFP construct, followed by single-cell cloning to establish the Ramos$^{CBR-GFP}$ NALM6$^{CBR-GFP}$, U937$^{CBR-GFP}$, and A20$^{CBR-GFP}$ cell lines.

The 9523 murine acute promyelocytic leukemia (APL) cells were previously generated in our laboratory by knocking in the human PML-RARa cDNA into the 5′ regulatory sequence of the cathepsin G locus, which produces a high-penetrance APL phenotype in 90% of mice[37]. Tumor cells were subsequently immortalized by serial propagation in vitro. Murine CD19 expression was added to 9523 cells by lentiviral gene transfer, followed by single cell cloning to establish a pure cell population of 9523$^{CD19}$.

**In vitro expansion assay.** Ramos$^{CBR-GFP}$ ($0.5 \times 10^6$) and UCART19 ($1 \times 10^6$) were plated in a 24-well plate in RPMI complete media with no cytokine supplementation. rhIL-7 or rhIL-7-hyFc was added to the culture to a final concentration of 10 ng/ml, 100 ng/ml, or 1000 ng/mg. Seven days later, cells were harvested and counted, and flow cytometry was performed to ensure no residual tumor cells remained in the culture. CAR T cells were then re-plated at a 2:1 effector to target (E:T) ratio in fresh media containing the same concentrations of rhIL-7 or rhIL-7-hyFc as the initial culture. On day 14 CAR T cells were harvested and analyzed by flow cytometry and single-cell cytokine analysis.

**Flow cytometry.** Cell lines and T cells from in vitro experiments were directly transferred into flow cytometry tubes and washed once prior to staining with fluorescently labeled antibodies for 15 min at room temperature. For peripheral blood evaluation, 50 µl was added to 2 ml of RBC lysis buffer in flow cytometry tubes and incubated for 10 min at room temperature, followed by a single wash and addition of Fc block (BioLegend #101320), then antibody staining. Spleens were macerated over a 70 um cell strainer using the piston from a 3 ml syringe to create single cell suspensions, after which 1-2e6 cells were transferred to flow cytometry tubes and washed once prior to Fc block and antibody staining.

All samples were run on either an Attune NxT Flow Cytometer (Thermo Fisher Scientific) or a ZE5 (Yeti) cytometer (Bio-Rad). Data analysis was performed using FlowJo v10.6.1 (Tree Star Inc.) or FCS Express v7.08.0018 (De Novo Software). The following reagents were used to evaluate human CAR T cell phenotype (all antibodies are against human antigens): CD34-PE (Beckman Coulter, IM1459U, dilution 1:30), LIVE/DEAD fixable yellow (Thermo Fisher Scientific, L34967, dilution 1:500), Annexin V-APC (BD Biosciences, 550474, dilution 1:10), Ki-67-APC (eBioscience, 17-5698-82, dilution 1:10), CD4-FITC (BioLegend, 300506, dilution 1:200), CD8-BV421 (BioLegend, 301036, dilution 1:80). For UCART33 we

used fluorescently labeled CD33 protein (Sino Biological, 12238-H05H-100, dilution 1:500) to detect CAR T cells, as this construct did not contain the human CD34 tag. Human T cell memory phenotype was evaluated using the following antibodies: CD45RO-BV421 (BioLegend, 304224, dilution 1:100), CD45RA-APC/Cy7 (Biolegend, 304128, dilution 1:100), CCR7-AF647 (BioLegend, 353218, dilution 1:100), CD3-BV786 (BioLegend, 317330, dilution 1:100), CD4-BV650 (BioLegend, 317436, dilution 1:100), CD8-AF700 (BioLegend, 300922, dilution 1:400).

Evaluation of murine peripheral blood was performed using the following antibodies (all against mouse antigens): CD45.1-PE/Cy7 (eBioscience, 25-0453-82, dilution 1:200), CD45.2-APC/ef780 (eBioscience, 47-0454-82, dilution 1:100), CD19-APC (BioLegend, 115512, dilution 1:400), Gr-1-BV605 (BioLegend, 108440, dilution 1:500), CD3-PE (BioLegend, 100206, dilution 1:100), and LIVE/DEAD fixable yellow (Thermo Fisher Scientific, L34967, dilution 1:500).

Detailed phenotyping of murine peripheral blood was performed using the following antibodies (all against mouse antigens): CD4-BUV395 (BD, 563790, dilution 1:400), LIVE/DEAD fixable blue (Thermo Fisher Scientific, L23105, dilution 1:500), CD11b-BUV661 (BD, 565080, dilution 1:500), B220-BUV737 (BD, 612838, dilution 1:500), CD44-BV421 (BD, 563970, dilution 1:500), CD8-BV510 (BioLegend, 100752, dilution 1:500), CD45.2-BV605 (BioLegend, 109841, dilution 1:500), Ly6G-BV650 (BioLegend, 127641, dilution 1:500), CD16/32-BV711 (BioLegend, 101337, dilution 1:500), CD49b-BV785 (BD, 740895, dilution 1:500), CD19-PerCP/C5.5 (BioLegend, 152405, dilution 1:500), Ly6C-PE-CF594 (BD, 562728, dilution 1:1000), CD45.1-PE/Cy5 (eBioscience, 15-0453-82, dilution 1:1000), CD127-PE/Cy7 (eBioscience, 25-1273-82, dilution 1:1000), CD3-APC (BD, 553066, dilution 1:500), CD62L-APC/Cy7 (BioLegend, 104428, dilution 1:500).

**Single-cell cytokine profiling of UCART19.** Single-cell cytokine production was measured using an IsoLight System (IsoPlexis). UCART19 cells were labeled with Violet stain A according to the manufacturer's protocol. Cells were incubated with targets at a 1:1 E:T ratio for 20 h. After stimulation, residual target cells were depleted with CD19 biotin microbeads using the Miltentyi Automacs according to the manufacturer's instructions. Cells were stained Alexa Fluor 647-conjugated anti-human CD8 (IsoPlexis) at room temperature for 20 min and loaded onto an IsoCode chip (IsoPlexis). Each IsoCode chip contains ~12,000 microchambers pre-patterned with a full copy of 32-plex antibody array including Effector: Granzyme B, TNFα, IFN-γ, MIP1α, Perforin, TNFβ; Stimulatory: GM-CSF, IL-2, IL-5, IL-7, IL-8, IL-9, IL-12, IL-15, IL- 21; Chemo-attractive: CCL11, IP-10, MIP-1β, RANTES; Regulatory: IL-4, IL-10, IL-13, IL-22, sCD137, sCD40L, TGFβ1; Inflammatory: IL-6, IL-17A, IL-17F, MCP-1, MCP-4, IL-1β. The polyfunctional profile (2+ proteins per cell) of single cells was evaluated by IsoSpeak software version 2.7.0.0.

**Mice.** All mice used in these studies were aged 6–12-week-old males purchased from Jackson Laboratories (Bar Harbor, ME): NOD-SCID-IL2Rγ$^{-/-}$ (NSG) (#005557), C57BL/6J (#000664), BALB/cJ (#000651), B6.SJL-Ptprca$^a$ Pepc$^b$/BoyJ (B6-CD45.1) (#002014), CByJ.SJL(B6)-Ptprca$^a$/J (Balb/c-CD45.1) (#006584). All animal experiments were performed according to an animal protocol approved by the Institutional Animal Care and Use Committee at Washington University School of Medicine. All experimental mice were co-housed within specific pathogen free facilities at Washington University School of Medicine and maintained on ad libitum water and standard chow (LabDiet 5053; Lab Supply, Fort Worth, TX), with a 12 h light/dark cycle and a temperature range of 68–74°F with 40–60% humidity. Mice were euthanized if they exhibited signs of illness or discomfort (tumor growth ≥2 cm, weight loss ≥20%, hind limb paralysis, lethargy, hunched posture), using carbon dioxide asphyxiation followed by cervical dislocation.

**In vivo mouse xenograft experiments.** NSG mice were injected intravenously (IV) via the lateral tail vein with $5 \times 10^5$ Ramos$^{CBR-GFP}$ or $5 \times 10^4$ U937$^{CBR-GFP}$ on day −4. On day 0, $1 \times 10^6$ UCART cells were injected IV. Mice received 10 mg/kg rhIL-7-hyFc subcutaneously (SC) on days +1, +15, and +29.

To track tumor growth in vivo, mice were injected intraperitoneally with 50 µg/g D-luciferin (Goldbio, eLUCNA), and bioluminescence was measured using an AMI HT optical imaging system (Spectral Instruments). Images were analyzed using Aura 4.0 In Vivo Imaging Software (Spectral Instruments).

**In vivo immunocompetent mouse experiments.** C57BL/6 mice were injected IV with $5 \times 10^4$ 9523$^{CD19}$ cells on day −3, followed by intraperitoneal (IP) injection of cyclophosphamide (Cayman, 13849) 250 mg/kg on day −1, then injected IV with $6 \times 10^6$ UTD or mCART19 (manufactured from CD45.1+ C57BL/6 mice) cells on day 0. Mice were then given 10 mg/kg rhIL-7-hyFc SC on days +1, +15, and +29. Mice were bled via facial vein every 4–6 days, and total white blood cell (WBC), hemoglobin, and platelet counts were measured using a Hemavet 950 analyzer (Drew Scientific). Absolute numbers of circulating leukocyte subsets were calculated by multiplying the WBC counts by the frequency of each cell type as measured by flow cytometry.

For in vivo mCART19 dose titration, Balb/c mice were injected IV with $1 \times 10^6$ A20 cells on day −4, followed by cyclophosphamide IP on day −1, then different doses of mCART19 (manufactured from CD45.1+ Balb/c mice) on day 0, followed by 10 mg/kg rhIL-7-hyFc SC on days +1, +15, and +29. Serial WBC counts and leukocyte subsets were measured as above.

To evaluate the anti-tumor activity of mCART19, Balb/c mice were irradiated with 600 cGy on day −8, followed by injection with either $5 \times 10^5$ A20$^{CBR-GFP}$ cells IV, or $1 \times 10^6$ A20$^{CBR-GFP}$ cells SC on day −7. Mice were then given IV injections of $5 \times 10^5$ mCART19 on day 0 and 10 mg/kg rhIL-7-hyFc SC on days +1 and +15. Tumor burden was followed by bioluminescent imaging as above.

To harvest mCART19 after rhIL-7-hyFc treatment in vivo, Balb/c mice were given cyclophosphamide 250 mg/kg IP on day −1 followed by $5 \times 10^5$ mCART19 cells IV on day 0. Mice were euthanized on day 7 and day 14 after mCART19, and spleens were harvested and macerated into single cell suspension using a 40 µm nylon cell strainer. Cells were then counted and analyzed by flow cytometry to assess leukocyte subsets. Adoptively transferred CD45.1+ cells from the spleens were purified using CD45.1-biotin (BioLegend, 110703) and anti-biotin microbeads (Miltenyi, 130-090-485), using the AutoMACS per manufacturer protocol.

**In vitro UCART19 cytotoxicity assay.** NALM6$^{CBR-GFP}$ or Ramos$^{CBR-GFP}$ target cells were seeded at a density of 25,000 cells per well in a 96-well plate in RPMI complete media. UCART19 cells were purified from the spleens of NSG mice on day +28 by labeling with CD34-PE antibody (Beckman Coulter, IM1459U) followed by anti-PE microbeads (Miltenyi, 130-048-801) and selecting bead-bound cells on the AutoMACS. Cryopreserved UCART19 input cells from Day 0 were used as a control. UCART19 cells were co-cultured with target cells at E:T ratios ranging from 4:1 to 1:4 at 37 °C for 20 h. Absolute cell counts of viable target cells were quantified by flow cytometry using 7-aminoactinomycin D and GFP.

**In vitro mCART19 cytotoxicity assay.** A20$^{CBR-GFP}$ target cells were seeded at a density of 50,000 cells per well in a black 96 well plate in RPMI complete media. mCART19 cells purified from spleens were co-cultured with target cells at E:T ratios ranging from 4:1 to 1:4 at 37 °C for 48 h. Residual tumor cells were quantified by bioluminescent imaging of the plate after adding 125ug/ml of D-luciferin (Goldbio, eLUCNA) using an AMI HT optical imaging system (Spectral Instruments). Percent cell lysis was calculated as follows: $[1 - (BLI^{treated}/BLI^{untreated})] \times 100$. All data points are technical duplicates.

**In vitro mCART19 intracellular cytokine staining.** In all, $2 \times 10^5$ mCART19 effector cells and $1 \times 10^6$ A20 target cells were co-cultured for 4 h at 37 °C in the presence of brefeldin A (BD Biosciences, 555029). Cells were then harvested and stained with LIVE/DEAD fixable yellow (Thermo Fisher Scientific, L34967) and CD3-APC/Cy7 (BioLegend, 100221), then treated with FIX & PERM cell permeabilization kit (ThermoFisher Scientific, GAS003) with the following antibodies against mouse cytokines: IFNg-PE (BioLegend, 505807), TNFa-APC (BioLegend, 506307).

**Single-cell RNA sequencing.** For xenograft tumor modeling in vivo, we injected NSG mice with $5 \times 10^5$ Ramos$^{CBR-GFP}$ cells 4 days before UCART19 ($1 \times 10^6$ cells) infusion. Mice were treated with rhIL-7-hyFc (10 mg/kg SC) or vehicle on days +1, and +15 post UCART19 infusion. On day +14 and day +28, mice were sacrificed and CAR-T cells were enriched from the spleen using CD34+ magnetic enrichment (Miltenyi). For the rhIL-7-hyFc treated groups ("week2" and "week4"), cells obtained from five mice per group were pooled on day +14 and day +28. For the Vehicle on control groups ("C1" and "C2"), cells were pooled from five mice per group on day +14. Additionally, UCART19 cells were harvested prior to infusion into mice on the day of UCART19 treatment ("Input"). Enriched CAR-T cells were resuspended in PBS containing 0.04% BSA, at a concentration of $1 \times 10^6$/ml, and submitted to the McDonnell Genome Institute for 3′ scRNA-seq. Utilizing the 10× Genomics Chromium Single Cell 3'v3 Library Kit and Chromium instrument, approximately 16,500–20,000 cells were partitioned into nanoliter droplets to achieve single-cell resolution for a maximum of 10,000 individual cells per sample. The resulting cDNA was tagged with a common 16nt cell barcode and 10nt Unique Molecular Identifier during the RT reaction. Full-length cDNA from poly-A mRNA transcripts was enzymatically fragmented and size selected to optimize the cDNA amplicon size (approximately 400 bp) for library construction (10x Genomics). The concentration of the 10x single cell library was accurately determined through qPCR (Kapa Biosystems) to produce cluster counts appropriate for the HiSeq 4000 or NovaSeq 6000 platform (Illumina). 26 × 98 bp sequence data were generated targeting 50K read pairs/cell, which provided digital gene expression profiles for each individual cell. Samples were aligned to the human and mouse genome (refdata-gex-GRCh38_and_mm10-2020-A) and demultiplexing, barcode processing and alignment were performed using CellRanger Single-Cell Software Suite v4.0.0. Cells were filtered using EmptyDrops[40] and further filtered to only maintain cells with <20% human mitochondrial DNA content and with a minimum of at least 200 and maximum of 20,000 genes expressed. Finally, to address multiplets of human and mouse cells, cells with <95% human mapped reads were filtered out. After filtering, <50 cells were maintained from the control arms (C1 and C2) and these samples were removed from downstream analyses. All single-cell data across all samples were merged together using Seurat merge and anchor functions.

**Statistical analysis**. Data plots were generated using GraphPad Prism 9.1.0 (La Jolla, CA). The determination of sample size and data analysis for this study followed the general guideline for animal studies[38]. The distributions of time-to-death were described using Kaplan–Meier product limit method and compared by the generalized Wilcoxon test which is less sensitive to the assumption of proportional hazards. All the other in vivo data were summarized using means and standard deviations. The differences were compared using two-sample Student's *t* test, one-way analysis of variance, or linear mixed model for repeated measurement data as appropriate, followed by post hoc multiple comparisons for between-group differences of interest. Based on the law of diminishing returns, Mead recommended that a pilot study with a sample size of 10–20 subjects would be adequate to estimate preliminary information[39]. The normality of data was assessed graphically using residuals and logarithm transformation was performed as necessary to better satisfy the normality and homoscedasticity assumptions. All analyses were two-sided and significance was set at a *p* value of 0.05. The statistical analyses were performed using SAS 9.4 (SAS Institutes, Cary, NC).

**Reporting summary**. Further information on research design is available in the Nature Research Reporting Summary linked to this article.

## Data availability

All data generated from this study are available within the paper and its Supplementary Information. Source data are provided with this paper. Human and mouse reference genomes (refdata-gex-GRCh38-and-mm10-2020-A) required for CellRanger can be downloaded from 10× Genomics (https://support.10xgenomics.com/single-cell-gene-expression/software/downloads/latest). The scRNA-seq data generated in this study has been deposited into the NCBI sequence read archive (SRA) database under BioProject accession PRJNA789884.

## Code availability

Custom code was not generated for this manuscript. The following publicly available pipelines were utilized for analysis: standard preprocessing and quality control of scRNA-seq data was based on Seurat Guided tutorials (https://satijalab.org/seurat/articles/pbmc3k_tutorial.html), splicing related analysis were performed by sjcount (https://github.com/pervouchine/sjcount), and monocle trajectory analysis was performed using standard tutorials (https://cole-trapnell-lab.github.io/monocle3/docs/trajectories/).

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

## Acknowledgements

The authors thank Julia Hollaway, Susan Gladney, Emily Street, Nicholas Wallace, and Kevin Kowal for assistance with animal care and experimental procedures and Sara Ferrando-Martinez and Alexandra Wolfarth for their critical review of the manuscript. We also thank the Siteman Flow Cytometry Core and the McDonnell Genome Institute at Washington University for providing technical assistance and equipment, and the Division of Comparative Medicine at Washington University for their excellent animal care. M.Y.K. was supported by an Alex's Lemonade Stand Foundation/Northwestern Mutual Young Investigator Award, a Washington University SPORE in Leukemia Developmental Research Award, and a Dean's Scholars Award from the Washington University Division of Physician-Scientists, which is funded by a Burroughs Wellcome Fund Physician-Scientist Institutional Award. M.P.R. was supported by an NCI Research Specialist Award (R50 CA211466). J.O. was supported by the International Myeloma Society and Paula and Roger Riney Foundation Translational Research Grant. J.F.D. was supported by an NCI Outstanding Investigator Award (R35 CA210084), an NIH P50 CA171963, and a Children's Discovery Institute Award.

## Author contributions

M.Y.K. designed and performed experiments, analyzed data, and wrote the manuscript. R.J. analyzed the single-cell RNA-seq data. J.M.D., J.R., M.P.R., J.O., K.W.S., K.M.K., and A.J.C. performed experiments and analyzed data. F.G. performed the statistical analyses. B.H.L. provided technical advice and reagent. M.L.C. supervised the project, designed the research, performed experiments, analyzed data, and wrote the manuscript. J.F.D. supervised the project, designed the research, and reviewed and edited the manuscript.

## Competing interests

M.Y.K., K.W.S., J.O., B.H.L., M.L.C., and J.F.D. are creators/inventors of a patent on the use of IL-7 to enhance CAR T cell function. B.H.L. is currently employed by NeoImmuneTech, Inc. M.L.C. is currently employed by and has equity ownership in Wugen. J.F.D. receives research funding from Amphivena Therapeutics, NeoImmuneTech, Macrogenics, Incyte, Bioline Rx; has equity ownership in Magenta Therapeutics, Wugen; consults for Incyte, RiverVest Venture Partners; and is a board member for Cellworks Group, Inc., RiverVest Venture Partners, Magenta Therapeutics. The remaining authors declare no competing interests.
