## [Peer Review File · Nature Communications]

A long-acting interleukin-7, rhIL-7-hyFc, enhances CAR T cell expansion, persistence and anti-tumor activityREVIEWER COMMENTS

Reviewer #1 (Remarks to the Author):

Dear Editor,

We recommend accepting the manuscript "A long-acting interleukin-7, rhIL-7-hyFc, enhances CAR-T expansion, persistence and anti-tumor activity in vivo" by Cooper, et. al. for publication in Nature Communications with the revisions outlined below. We agree with the authors that despite remarkable advances in the use of CAR-T therapies, challenges exist, including optimization of therapy for some patients. We anticipate the readership will find the work thought provoking and interesting given the results of interleukin-7 (IL-7) fused with hybrid Fc (rhIL-7-hyFc) on CAR-T cells.

Major Findings: rhIL-7-hyFc dramatically enhanced UCART expansion, persistence and anti-tumor efficacy in vivo, and promoted the transcription of genes associated with T cell proliferation and survival, resulting in significantly prolonged survival of mice in CD19+ lymphoma and CD33+ AML xenograft models.

Using a cancer model that is "less than responsive" to other therapeutics, Ramos (an aggressive B lymphoma model) the authors showed that NSG mice treated with Universal CAR-T cells against CD19+ lymphoma (UCART19) alone survived a median of 29 days, compared to untreated RamosCBR-GFP bearing NSG mice, 18 days. As stated by the authors, it is indeed, "remarkable that 100% of RamosCBR-GFP bearing mice treated with UCART19 and rhIL-7-hyFc were alive at 80 days (Figure 2B; UCART19+rhIL-7-hyFc vs. UCART19, $p=0.018$), with no mouse showing clinical signs of xenogeneic GVHD." The efficacy of rhIL-7-hyFc on CAR-T expansion was supported by studies performed using an orthogonal AML-specific CAR-T in an NSG model of acute myeloid leukemia. The results of a second model, UCART33 on CD33+ AML xenogeneic mice, showed similarly impressive promotion of CAR-T activity. Quantitative flow cytometric analyses of blood revealed logarithmic expansion of UCART19 or UCART33 cells in the blood of mice receiving rhIL-7-hyFc. UCART cells of mice receiving rhIL-7-hyFc were more cytotoxic ex-vivo. Overall, the authors submitted an interesting set of data to show the use of rhIL-7-hyFc as a viable adjuvant for enhancing CAR-T cell efficacy and tumor killing in vivo.

The manuscript lacks clarity and details that should be corrected. It is also our recommendation that the authors provide greater detail to the descriptions within the figure legends which were confusing and it seems likely the readership rely on for a quick assessment and analysis of the presented data. A major deficiency was the lack of control and experimental animal group sizes within figure legends and methods. Including this information is necessary for reviewer/reader to draw their own conclusions and access the design and interpretation of the data presented. More specifically, it is our belief can be improved should the following deficiencies, detailed below, be adequately addressed.

Major Points

The figure legends should contain descriptions that allow the reader to succinctly assess the experimental design and interpret the data. The majority of corrections below are related to the figure legends.

- We were unable to find the sample size of each animal control and test groups within the figure legends, Methods (In Vivo Efficacy) section or body of the manuscript. As stated above, the information is important for the reader to assess the study.
- Figure 1H and 1I are difficult to interpret since the figure legends and text lack description of the meaning of the gradient orange sections (H) and the difference between green/purple/aqua/yellow sections (I) of the bar graphs.

Minor Points

- It is our opinion that Figure legends are easier to follow if the alphabetic label proceeds the description of the sub-image. For example: (A) rhIL-7-hyFc is an engineered 341 IL-7 homodimer fused to IgD and IgG4 elements (hyFc[©]), promoting in vivo stability and reducing complement 342 activation. Opposed to the current format used: rhIL-7-hyFc is an engineered 341 IL-7

homodimer fused to IgD and IgG4 elements (hyFc[©]), promoting in vivo stability and reducing complement 342 activation (A).

- State that E:T is Effector: Target ratio before Figure 4, which is the first time it is written in full form but not the first time you observe E:T. Readers may not intuitively know what E:T ratio is and need to see it written out before the abbreviation is used frequently.
- E:T ratio should be expanded to Effector:Target ratio the first time used in Methods. First time we saw it as Effector:Target is in "In Vitro Expansion Assay" section which is not the first time E:T was used in the Methods.
- Line 393: Day 0 (Input), missing the closing parenthesis
- Line 553: RPMI 1640 written as RPMI1 640 in the text
- Line 590: ("week2" and "week"), second week is missing a number.
- Line 593: "prior" is misspelled

Figure 1

1. Legend 1C is not referenced or described within the Figure legend. The text reads from (B) to (D) in the figure description.
2. From Figure 1G, it is not readily apparent if the profile shown is for UCAR19 control or UCAR19 with rhIL-1-hyFc. This should be clearly indicated in the figure legend.

Figure 2

1. Figure 2B. The findings are intriguing, however, since the figure legend does not indicate the number of animals per group, the conclusiveness and impact of the findings is less compelling. A scan of the Methods also failed to identify this information.
2. Figure 2C. The values for (*) and (***) should be clear in the figure legend. Presumably, **** is $p < 0.0001$ and * is $p < 0.05$ but it is not stated. Please clarify.
3. The data for Figure 2D, grouped with 2C, which is acceptable, however, there is no information given regarding the left hand side of image 2D. It is not clear what the Min, Max values refer to without having to search the body of the manuscript.
4. Figure 2E. Was the blood pooled and then analyzed? If yes, it should be clearly stated and the information included as to how many animals per group were assessed.
5. Figure 2G and 2H. Please indicate how many mice per point. N=1 for each point? It is not stated and confuses the interpretation of the study.
6. Figure 2H. Analysis of the findings from Figure 2H were not located within the body of the manuscript. The authors should discuss their analysis of the data points following the rechallenge (day +275). Are they concerned that after the rechallenge photon detection was similar to mice never treated with rhIL-7?

Figure 3

1. 3B does not state how many animals per group or per data point. Please clearly state what ** denotes in the figure legend.
2. 3C figure legend does not specify what the Min, Max values refer to on the right hand side of the image. The reader should be able to quickly locate the information in the figure legend to assess the data.
3. 3D, please state what **** denotes in the figure legend.
4. 3E, please state what ** denotes. Gaps in the description leads to a failure to convey key information such as how the data was collected other than stating it was a measure of CAR-T cells from peripheral blood. Point of fact, how were CAR-T cell numbers established?

Figure 4

- 4B and 4C, please state the values of *** and **** denote in the figure legend.

Figure 5

- Text references to figure 5E-F do not appear to align with the figure or figure legend. Does this mean the experiment or interpretation of this study similarly uncoupled?
- 5E, Exhaustion Score is not explained in the figure legend. It is explained in the body of the manuscript, however, since this is not a measurement that every reader will be familiar with it might be important to add detail such as Low Score=light blue, High score=purple or some similar connotation which should be considered. Is it sufficient to label it from 0 to 1 when a value of 0 may be preferable?

Reviewer #2 (Remarks to the Author):

Cooper et al show that a rhIL-7-hyFc construct administered at day 1, 15 and 29 enhances CAR-T cell survival and cytotoxicity in xenogeneic models of B cell leukemia and AML. Overall, the experiments are well performed and presented and the findings are unequivocal. Yet, the main caveat of the study is the xenogenic barrier as the beneficial effects of rhIL-7 in an IL-7 deficient host are not predictive for therapeutic effects in IL-7 replete humans. A syngeneic tumor model confirming enhanced CAR T cell efficacy upon IL-7 administration would have strengthened the study.

Minor comments:

A comparison to unmodified rh-IL-7 would have been nice.

Fig. 1E: it seems CD8 T cells preferentially expand upon rhIL-7-hyFc treatment. This could be discussed.

Fig. 2H: the proposed memory response upon tumor rechallenge by long-term surviving CARs is not convincing in these experiments

Fig. 3C: why was the threshold for BLI at d49 and d167 set to 5×10^4 (in contrast to earlier time points)?

Fig. 2F and 3E: CAR T cell numbers in PB seem to rise late, preferentially after the 3rd rhIL-7-hyFc treatment (when the tumor is already cleared). This could be discussed.

Fig. 5: very nice sc-RNA seq analysis; yet the transition to memory T cells is already shown by FACS and RNAseq data reveal only marginal additional information

Discussion: 1st sentence states „...target negative relapse“ but it´s probably meant „...target positive relapse“

Reviewer #3 (Remarks to the Author):

In this well-written and well-conducted study, the authors investigate the roles of rhIL-7-hyFc, in vitro and in vivo, on CAR T-cell expansion, persistence and anti-tumor effects. The current working hypothesis is that after encountering their target antigen, CAR-T cells differentiate into effector-like phenotypes, and may acquire an exhaustion phenotype, hindering their long-term persistence and anti-tumor effects. Here, the authors successfully demonstrate enhanced in vivo expansion and persistence along with superior antitumor clearance in mice treated with rhIL-7-hyFc using CAR T-cell targeting CD19 or CD33.

Major comments:

- The authors conclude from in vitro assays that rhIL-7-hyFc improves CAR T-cell function, based on increased CAR-T cell numbers after coculture with rhIL-7-hyFc and small differences in cytokine polyfunctionality indexes. An alternative hypothesis should be tested: that the increased number of CAR T-cells is primarily due to a survival advantage due to the anti-apoptotic effects of rhIL-7-hyFc, rather than an increase in their proliferative capacity per se. CFSE assay of CAR-T cells with/without CD19 after in vitro stimulation and Annexin-V, Ki-67 expression may provide important insights. Also needed is an in vitro cytotoxicity assay to assess anti-tumor function across a range of effector/target ratios comparing CAR-T alone to CAR-T cocultures with rhIL-7-hyFc.

- Figure 4 compares the cytotoxic effects of CAR-T cells having expanded in vivo, isolated from the mice's spleen at day +28, compared to "input" CAR-T cells. The authors showed higher cytotoxicity associated with day +28 CAR-T cells exposed to rhIL-7-hyFc. Again to firmly demonstrate increased functionality associated with rhIL-7-hyFc additional assays are needed: CAR-T cell proliferation (CFSE), cytokine production (isoplexis or FACS/intracellular staining) after in vitro stimulation.

- Limitations of xenograft NSG mice models should be emphasized. As reflected in Fig 2G, very high doses of CAR-T cells are needed in these models to achieve in vivo tumor clearance. This might be due – in part – to the lack of cytokine support from the myeloid compartment, of which NSG mice are deficient. Cytokine response to lymphodepletion in human (mostly produced by endogenous myeloid cells) is known to play a key role in anti-lymphoma effects in humans (Hirayama et al, Blood 2019). Humanized mouse models would be more relevant (see. Norelli et

al, Nature Medicine 2018). It is very unclear to me how the findings presented will translate in humans.

Minor comments:

- The authors should comment on rhIL-7-hyFc dosing in their mouse model (approximately 100-fold higher than the dose used in humans based on NCT02860715). Toxicity will be a concern at high doses.
- Figure 1H is difficult to interpret without a legend specifying the color coding. Please edit as appropriate.
- Figure 1J: again difficult to distinguish visually major differences across the three groups. PCA or t-SNE analysis (or other dimensionality-reduction approaches) could be useful to show which analytes accounts for most of the variance in polyfunctionality.
- Regarding the statement p7 to overcome manufacturing issues with CAR-T cell target dose not being reached. I think this is weak rationale, the percentage of manufacturing failures across products is currently low, e.g., with Yescarta 97% of products have been reported "on-spec". Insufficient dose is only one of many causes of manufacturing failures, poor cell viability being another important cause. See <https://www.biopharma-reporter.com/Article/2019/04/01/Gilead-produces-97-on-spec-Yescarta> A better angle could be to use rhIL-7-hyFc as a CAR-T dose-sparing strategy to allow subsequent repeat infusions in patients.
- To demonstrate that rhIL-7-hyFc can improve persistence, beyond initial peak expansion, it would be useful to compare rhIL-7-hyFc-treated mice to vehicle-treated mice receiving an intermediate/high dose of CAR-T cells, with the goal of obtaining comparable CAR-T cell peak expansion, and to compare the CAR-T cell AUC or CAR-T numbers at selected timepoints.
- Authors report p11 somewhat provocative data with decreased "genotypic" exhaustion (based on the author's own scoring metric) across time. Can the authors put forth hypotheses and limitations to explain this?
- Figure 5: did the authors look at TOX and Ki-67 expression specifically?
- What is the post-in vivo expansion phenotype of CAR-T cells treated with higher dose without rhIL-7-hyFc? Differences compared to rhIL-7-hyFc-treated mice?
- Looking at the data showed in Fig 5C, cluster 0 and 1 seem connected to multiple clusters, not only Cluster 4 as reported in the manuscript.
- Did the authors experience difficulties with xenoGVHD prompting them to knock-out the endogenous TCR? If so, this should be clearly mentioned and additional data should be provided as Supplementary material.
- Discussion: I agree with the authors that immunogenicity may be worsened with rhIL-7-hyFc in the allogeneic setting. Importantly, this could also be a concern in the autologous setting, a situation in which CD8+-mediated anti-CAR immune responses have been detected (see Gauthier J et al Blood 2020). This should be mentioned in the discussion.
- Methods: a section should describe the murine models in more details
- Statistical testing was not always performed (e.g. in vitro assays in Fig 1). Was there a rationale to decide whether or not to perform hypothesis testing?
- "Based on the law of diminishing returns, Mead recommended that a degree of freedom (DF) of 10-20 associated with error term in an ANOVA will be adequate for a pilot study to estimate preliminary information [42]." is this statement appropriate or copy-pasting error?

Jordan Gauthier, MD, MSc

REVIEWER COMMENTS

Reviewer #1 (Remarks to the Author):

Dear Editor,

We recommend accepting the manuscript “A long-acting interleukin-7, rhIL-7-hyFc, enhances CAR-T expansion, persistence and anti-tumor activity in vivo” by Cooper, et. al. for publication in Nature Communications with the revisions outlined below. We agree with the authors that despite remarkable advances in the use of CAR-T therapies, challenges exist, including optimization of therapy for some patients. We anticipate the readership will find the work thought provoking and interesting given the results of interleukin-7 (IL-7) fused with hybrid Fc (rhIL-7-hyFc) on CAR-T cells.

Major Findings: rhIL-7-hyFc dramatically enhanced UCART expansion, persistence and anti-tumor efficacy in vivo, and promoted the transcription of genes associated with T cell proliferation and survival, resulting in significantly prolonged survival of mice in CD19+ lymphoma and CD33+ AML xenograft models.

Using a cancer model that is “less than responsive” to other therapeutics, Ramos (an aggressive B lymphoma model) the authors showed that NSG mice treated with Universal CAR-T cells against CD19+ lymphoma (UCART19) alone survived a median of 29 days, compared to untreated RamosCBR-GFP bearing NSG mice, 18 days. As stated by the authors, it is indeed, “remarkable that “100% of RamosCBR-GFP bearing mice treated with UCART19 and rhIL-7-hyFc were alive at 80 days (Figure 2B; UCART19+rhIL-7-hyFc vs. UCART19, $p=0.018$), with no mouse showing clinical signs of xenogeneic GVHD.” The efficacy of rhIL-7-hyFc on CAR-T expansion was supported by studies performed using an orthogonal AML-specific CAR-T in an NSG model of acute myeloid leukemia. The results of a second model, UCART33 on CD33+ AML xenogeneic mice, showed similarly impressive promotion of CAR-T activity. Quantitative flow cytometric analyses of blood revealed logarithmic expansion of UCART19 or UCART33 cells in the blood of mice receiving rhIL-7-hyFc. UCART cells of mice receiving rhIL-7-hyFc were more cytotoxic ex-vivo. Overall, the authors submitted an interesting set of data to show the use of rhIL-7-hyFc as a viable adjuvant for enhancing CAR-T cell efficacy and tumor killing in vivo.

The manuscript lacks clarity and details that should be corrected. It is also our recommendation that the authors provide greater detail to the descriptions within the figure legends which were confusing and it seems likely the readership rely on for a quick assessment and analysis of the presented data. A major deficiency was the lack of control and experimental animal group sizes within figure legends and methods. Including this information is necessary for reviewer/reader to draw their own conclusions and access the design and interpretation of the data presented. More specifically, it is our belief can be improved should the following deficiencies, detailed below, be adequately addressed.

Major Points

The figure legends should contain descriptions that allow the reader to succinctly assess the experimental design and interpret the data. The majority of corrections below are related to the figure legends.

- We were unable to find the sample size of each animal control and test groups within the figure legends, Methods (In Vivo Efficacy) section or body of the manuscript. As stated above, the information is important for the reader to assess the study.*

Response: We apologize for this deficiency, and thank the reviewer for bringing this to our attention. Figure legends have been updated to add sample size to each experiment.

• *Figure 1H and 1I are difficult to interpret since the figure legends and text lack description of the meaning of the gradient orange sections (H) and the difference between green/purple/aqua/yellow sections (I) of the bar graphs.*

Response: We thank the reviewer for bringing this ambiguity in the figure to our attention. The colors of the bar graph correspond to the cytokine categories in Figure 1G of the original manuscript. We have revised the figure to add a simplified legend directly to the graph for clarity.

Minor Points

• *It is our opinion that Figure legends are easier to follow if the alphabetic label proceeds the description of the sub-image. For example: (A) rhIL-7-hyFc is an engineered 341 IL-7 homodimer fused to IgD and IgG4 elements (hyFc©), promoting in vivo stability and reducing complement 342 activation. Opposed to the current format used: rhIL-7-hyFc is an engineered 341 IL-7 homodimer fused to IgD and IgG4 elements (hyFc©), promoting in vivo stability and reducing complement 342 activation (A).*

Response: The figure legends have been updated so that the label precedes the description, as the reviewer suggests.

• *State that E:T is Effector: Target ratio before Figure 4, which is the first time it is written in full form but not the first time you observe E:T. Readers may not intuitively know what E:T ratio is and need to see it written out before the abbreviation is used frequently.*

Response: We have updated the legend and the text to clarify the meaning of E:T ratio the first time it is used.

• *E:T ratio should be expanded to Effector:Target ratio the first time used in Methods. First time we saw it as Effector:Target is in “In Vitro Expansion Assay” section which is not the first time E:T was used in the Methods.*

Response: We have updated the Methods to clarify the meaning of E:T ratio the first time it is used.

- *Line 393: Day 0 (Input), missing the closing parenthesis*
- *Line 553: RPMI 1640 written as RPMI1 640 in the text*
- *Line 590: (“week2” and “week”), second week is missing a number.*
- *Line 593: “prior” is misspelled*

Response: We thank the reviewer for being reviewing our manuscript so thoroughly. These errors have been corrected.

Figure1

1. *Legend 1C is not referenced or described within the Figure legend. The text reads from (B) to (D) in the figure description.*

Response: This has been corrected.

2. From Figure 1G, it is not readily apparent if the profile shown is for UCAR19 control or UCAR19 with rhIL-1-hyFc. This should be clearly indicated in the figure legend.

Response: Figure 1G is a schema of the different cytokines included in the analysis. We have moved this schema to Supplemental Figure 2 to avoid ambiguity in this regard. The figure legend has also been updated to clarify this point.

Figure 2

1. Figure 2B. The findings are intriguing, however, since the figure legend does not indicate the number of animals per group, the conclusiveness and impact of the findings is less compelling. A scan of the Methods also failed to identify this information.

Response: We apologize for this omission. The figures and legends have been updated to include the number of animals per group.

2. Figure 2C. The values for (*) and (****) should be clear in the figure legend. Presumably, **** is $p < 0.0001$ and * is $p < 0.05$ but it is not stated. Please clarify.

Response: We apologize for this omission. The values have been updated in the legend as follows: * $p \leq 0.05$, ** $p \leq 0.01$, *** $p \leq 0.001$, **** $p \leq 0.0001$.

3. The data for Figure 2D, grouped with 2C, which is acceptable, however, there is no information given regarding the left-hand side of image 2D. It is not clear what the Min, Max values refer to without having to search the body of the manuscript.

Response: Figure 2D has been updated to include a color scale to clearly define that the values represent bioluminescent signal intensity.

4. Figure 2E. Was the blood pooled and then analyzed? If yes, it should be clearly stated and the information included as to how many animals per group were assessed.

Response: Figure 2E shows representative FACS plots from two individual mice over time, while Figure 2F shows values from each mouse. The legend has been updated to reflect this.

5. Figure 2G and 2H. Please indicate how many mice per point. $N=1$ for each point? It is not stated and confuses the interpretation of the study.

Response: Figure 2G and 2H have been updated for clarity. Each group of mice receiving the same dose of UCART19 has been split into a separate plot. For the BLI images, each line represents one mouse. The figure legend has been updated to reflect this.

6. Figure 2H. Analysis of the findings from Figure 2H were not located within the body of the manuscript. The authors should discuss their analysis of the data points following the rechallenge (day +275). Are they concerned that after the rechallenge photon detection was similar to mice never treated with rhIL-7?

Response: We agree with the reviewer that most of the mice that received tumor rechallenge showed tumor progression to a similar degree as the control mice, although one mouse did achieve tumor clearance. We believe that this is a limitation of the NSG model, which may not support formation of long-lasting human memory T cells. This data has now been removed from the paper, and tumor rechallenge data from our immunocompetent mouse model has been

added as **Figure 5G** shown below, where the protective effect of prior CAR T cells with rhIL-7-hyFc is more evident.

Revised Figure 5G:

Figure 3

1. 3B does not state how many animals per group or per data point. Please clearly state what ** denotes in the figure legend.
2. 3C figure legend does not specify what the Min, Max values refer to on the right hand side of the image. The reader should be able to quickly locate the information in the figure legend to assess the data.
3. 3D, please state what **** denotes in the figure legend.
4. 3E, please state what ** denotes. Gaps in the description leads to a failure to convey key information such as how the data was collected other than stating it was a measure of CAR-T cells from peripheral blood. Point of fact, how were CAR-T cell numbers established?

Response: The figure legend has been updated to include the number of animals per group, and $** p \leq 0.01$, $***p \leq 0.001$, $**** p \leq 0.0001$ has been stated. CAR T cell numbers were established by quantitative flow cytometry analysis, and this has been added to the figure legend.

Figure 4

- 4B and 4C, please state the values of *** and **** denote in the figure legend.

Response: The values have been updated in the legend as follows: $***p \leq 0.001$, $**** p \leq 0.0001$.

Figure 5

- Text references to figure 5E-F do not appear to align with the figure or figure legend. Does this mean the experiment or interpretation of this study similarly uncoupled?

Response: We thank the reviewer for identifying this discrepancy and have rewritten both the figure legend and the relevant text to properly align these results.

- 5E, Exhaustion Score is not explained in the figure legend. It is explained in the body of the manuscript, however, since this is not a measurement that every reader will be familiar with it might be important to add detail such as Low Score=light blue, High score=purple or some

similar connotation which should be considered. Is it sufficient to label it from 0 to 1 when a value of 0 may be preferable?

Response: We appreciate this feedback and have accordingly expanded the figure legend to help guide the reader through this analysis, as follows:

“Downsampled objects (n=2,000 cells for each treatment group) colored by exhaustion score, calculated based on the expression of genes known to be involved in T cell exhaustion (*CTLA4*, *PDCD1*, *LAG3*, *HAVCR2*, *CD160*, *CD244*, and *TIGIT*). Dark purple indicates a higher exhaustion phenotype, and light blue indicates a less exhaustive phenotype.”

Reviewer #2 (Remarks to the Author):

Cooper et al show that a rhIL-7-hyFc construct administered at day 1, 15 and 29 enhances CAR-T cell survival and cytotoxicity in xenogeneic models of B cell leukemia and AML. Overall, the experiments are well performed and presented and the findings are unequivocal. Yet, the main caveat of the study is the xenogenic barrier as the beneficial effects of rhIL-7 in an IL-7 deficient host are not predictive for therapeutic effects in IL-7 replete humans. A syngeneic tumor model confirming enhanced CAR T cell efficacy upon IL-7 administration would have strengthened the study.

Response: We thank the reviewer for this insightful comment. Per the reviewer's suggestion, we have evaluated the effects of rhIL-7-hyFc on CAR T cells in immunocompetent mouse models with syngeneic tumor (Figures 4-7). These data support our conclusions that rhIL-7-hyFc enhances CAR T cell expansion, persistence and anti-tumor efficacy.

Minor comments:

A comparission to unmodified rh-IL-7 would have been nice.

Response: Per the reviewer's suggestion, we have conducted in vitro experiments comparing unmodified rhIL-7 to rhIL-7-hyFc in Figure 1D-F. We find that the effects of these two reagents on CAR T cells are nearly identical, as expected. The main benefit of rhIL-7-hyFc over unmodified rhIL-7 is the extended in vivo half-life, providing more durable effect. This has also been emphasized in the discussion.

Fig. 1E: it seems CD8 T cells preferentially expand upon rhIL-7-hyFc treatment. This could be discussed.

Response: We observed a high proportion of CD8+ T cells in the initial rhIL-7-hyFc expanded UCART19 cells in vitro, as the reviewer notes. However, this finding is not consistent across different T cell donors, across different timepoints in vivo, or across species (e.g. murine CAR T cells show a more significant increase in CD4+ T cells after rhIL-7-hyFc treatment). Therefore, we have thus refrained from discussing this in the manuscript.

Fig. 2H: the proposed memory response upon tumor rechallenge by long-term surviving CARs is not convincing in these experiments

Response: We agree with the reviewer that the tumor rechallenge data in long-term surviving NSG mice after UCART19+rhIL-7-hyFc is not robust, and have removed this from the manuscript. Instead, we have added tumor rechallenge data from our immunocompetent mouse model as **Figure 5G** shown below, where the protective effect of prior CAR T cells with rhIL-7-hyFc is more evident.

Revised Figure 5G:

Fig. 3C: why was the threshold for BLI at d49 and d167 set to 5×10^4 (in contrast to earlier time points)?

Response: Due to the low overall signal in the mice at the later timepoints, we had to increase the minimum threshold to be able to show the mice clearly. Below is what the images look like with a minimum of 0 as opposed to 5×10^4 at day 49. Changing the threshold parameters change the appearance of the images but do not affect the absolute BLI values, and thus we believe that this is a liberty we can take to emphasize the fact that all of the long-term surviving mice were tumor-free with very low BLI signal.

Fig. 2F and 3E: CAR T cell numbers in PB seem to rise late, preferentially after the 3rd rhIL-7-hyFc treatment (when the tumor is already cleared). This could be discussed.

Response: In the NSG model human CAR T cells logarithmically increase in response to rhIL-7-hyFc, and this is more prominent at the later timepoints (week 4-6). We do see higher CAR T cell numbers even after one dose of rhIL-7-hyFc, although we realize that this is difficult to ascertain based on the logarithmic plots that we showed in the manuscript, and thus we have broken down the values by week for the reviewer (see below; week 2 measurements were taken prior to the 2nd dose of rhIL-7-hyFc). Each dose of rhIL-7-hyFc has a cumulative effect, so that CAR T cell numbers peak after the third dose. However, the CAR T cells still require antigen recognition for expansion. A separate cohort of mice in the NSG+U937+CART33 experiment had received UCART33+rhIL-7-hyFc without tumor, and as you see there is minimal

CAR T cells detected in this group. This data has been added to **Figure 3E** in the revised manuscript.

Revised Figure 3E

We would also like to bring to the reviewer's attention that in the immunocompetent mouse model, we see increase of mCAR19 after the first dose of rhIL-7-hyFc, and these numbers do not increase further after the 2nd and 3rd doses (**Figures 4D and 6C**, reproduced below). These results are more in line with what we would expect to occur in IL-7 replete human subjects.

Revised Figures 4D and 6C:

These observations have been discussed per the reviewer's suggestion as follows:

“murine CAR T cell numbers in immunocompetent mice did not reach the astronomical levels seen with human CAR T cells in NSG mice after rhIL-7-hyFc. The NSG model provides proof-of-principle that this reagent is effective on human CAR T cells in vivo, but the lack of endogenous lymphocytes in these mice is likely responsible for the massive expansion of T cells, and in human subjects the kinetics of CAR T cell expansion is expected to be more akin what was seen in the immunocompetent mice”

Fig. 5: very nice sc-RNA seq analysis; yet the transition to memory T cells is already shown by FACS and RNAseq data reveal only marginal additional information

Response: We would respectfully disagree with the reviewer that the scRNA-seq data does not add additional information to this study. First, we believe that using multiple modalities to validate our findings is important for scientific rigor, and thus the correlation between the FACS and scRNA-seq results in itself is meaningful. Additionally, we were able to identify a subset of cells with an exhausted-like memory state, which was not evident in the FACS analysis. Furthermore, the scRNA-seq data allowed us to identify 15 unique T cell transcriptional profiles across treatment groups that we can track over time, and thus provides a much more comprehensive picture of the effects of rhIL-7-hyFc on CAR T cell phenotype.

Discussion: 1st sentence states „...target negative relapse“ but it's probably ment „...target positive relapse“

Response: We thank the reviewer for identifying this error and have corrected it in the revised manuscript.

Reviewer #3 (Remarks to the Author):

In this well-written and well-conducted study, the authors investigate the roles of rhIL-7-hyFc, in vitro and in vivo, on CAR T-cell expansion, persistence and anti-tumor effects. The current working hypothesis is that after encountering their target antigen, CAR-T cells differentiate into effector-like phenotypes, and may acquire an exhaustion phenotype, hindering their long-term persistence and anti-tumor effects. Here, the authors successfully demonstrate enhanced in vivo expansion and persistence along with superior antitumor clearance in mice treated with rhIL-7-hyFc using CAR T-cell targeting CD19 or CD33.

Major comments:

- The authors conclude from in vitro assays that rhIL-7-hyFc improves CAR T-cell function, based on increased CAR-T cell numbers after coculture with rhIL-7-hyFc and small differences in cytokine polyfunctionality indexes. An alternative hypothesis should be tested: that the increased number of CAR T-cells is primarily due to a survival advantage due to the anti-apoptotic effects of rhIL-7-hyFc, rather than an increase in their proliferative capacity per se. CFSE assay of CAR-T cells with/without CD19 after in vitro stimulation and Annexin-V, Ki-67 expression may provide important insights. Also needed is an in vitro cytotoxicity assay to assess anti-tumor function across a range of effector/target ratios comparing CAR-T alone to CAR-T cocultures with rhIL-7-hyFc.

Response: We thank the reviewer for these excellent suggestions to investigate the mechanism of rhIL-7-hyFc enhancement of CAR T cell proliferation and function. We have performed Annexin V and Ki-67 staining of UCART19 after in vitro expansion with and without rhIL-7-hyFc. While there was a slight decrease in apoptosis, the main driving factor of increased CAR T cell numbers was increased proliferation, as Ki-67 expression was markedly increased with rhIL-7-hyFc treatment. These findings have been updated in the revised paper in **Figure 1E**, reproduced below:

Revised Figure 1E

One of the primary limitations in comparing the function of CAR T cells with and without rhIL-7-hyFc treatment has been that without rhIL-7-hyFc the quantity of CAR T cells are often insufficient to do functional assays, as can be seen in the growth curves in Figure 1D. Unfortunately, due to this limitation we were unable to perform the CFSE or in vitro cytotoxicity assays that the reviewer suggests.

- Figure 4 compares the cytotoxic effects of CAR-T cells having expanded in vivo, isolated from the mice's spleen at day +28, compared to "input" CAR-T cells. The authors showed higher

cytotoxicity associated with day +28 CAR-T cells exposed to rhIL-7-hyFc. Again to firmly demonstrate increased functionality associated with rhIL-7-hyFc additional assays are needed: CAR-T cell proliferation (CFSE), cytokine production (isoplexis or FACS/intracellular staining) after in vitro stimulation.

Response: We agree with the reviewer that this was not a comprehensive assay of CAR T cell function. To address this limitation, we have added additional data on CAR T cell function from our immunocompetent mouse models, where we have the added benefit of being able to isolate enough mCART19 cells from the non-rhIL-7-hyFc treated mice, at least at an early time point (day 7). We show that rhIL-7-hyFc treatment in vivo decreases apoptosis (as measured by Annexin V binding), increases proliferation (as measured by Ki-67 expression), increases the cytotoxicity of mCART19, and increases cytokine production (as measured by production of IFN γ and TNF α). These findings have been updated in the revised paper in Figure 7D-G, reproduced below:

Revised Figure 7D-G

- Limitations of xenograft NSG mice models should be emphasized. As reflected in Fig 2G, very high doses of CAR-T cells are needed in these models to achieve in vivo tumor clearance. This might be due – in part – to the lack of cytokine support from the myeloid compartment, of which NSG mice are deficient. Cytokine response to lymphodepletion in human (mostly produced by endogeneous myeloid cells) is known to play a key role in anti-lymphoma effects in humans (Hirayama et al, Blood 2019). Humanized mouse models would be more relevant (see. Norelli et al, Nature Medicine 2018). It is very unclear to me how the findings presented will translate in humans.

Response: The reviewer is correct that xenograft NSG mouse models have many limitations, however this is still the standard model used for human CAR T cell studies. Our studies in the NSG model used TCR-deficient CAR T cells to at least avoid allogeneic and xenogenic GVHD, which can be a major confounding factor in interpreting tumor clearance and survival data. As the reviewer notes, humanized mouse models are better suited to capture the interactions between human myeloid cells and CAR T cells, and can serve as a model system to recapitulate toxicities such as cytokine release syndrome, in which myeloid cells play an important role. However, humanized mouse models also have limitations, particularly when studying lymphoid cells such as human T cells and NK cells, as these do not engraft robustly

nor function properly in the mouse host. Additionally, the humanized mouse model suggested by the reviewer uses NSGS mice that constitutively express human cytokines (SCF, GM-CSF, IL-3) to enhance myeloid engraftment, but these cytokines can also affect CAR T cells and confound results when trying to assess the effects of rhIL-7-hyFc. Thus, to better recapitulate the effects of rhIL-7-hyFc in humans, we used immunocompetent mouse models testing the effects of rhIL-7-hyFc on murine CAR T cells. These findings are presented in the revised paper in Figures 4-7, and we hope the reviewer finds this data informative for translation into human studies. We have also opened a clinical study (NCT05075603) to test the impact of rhIL-7-hyFc on CD19-targeting CAR T cells in humans, and will report the findings from this trial in a future paper.

Minor comments:

- *The authors should comment on rhIL-7-hyFc dosing in their mouse model (approximately 100-fold higher than the dose used in humans based on NCT02860715). Toxicity will be a concern at high doses.*

Response: We thank the reviewer for the opportunity to clarify our rationale behind the dose of rhIL-7-hyFc in our experiments. We chose a dose of 10mg/kg because we wanted to ensure that the mice received saturating doses of the drug. Lower doses would also likely have been effective, but we did not perform dose-finding studies of rhIL-7-hyFc as this was outside the scope of our paper. We would like to bring to the reviewer's attention that smaller animals have a higher metabolic rate, and thus higher drug doses are required in mice than in humans (ref: PMID 27057123). Therefore, a dose of 10mg/kg in mice is equivalent to 813ug/kg in humans. NCT02860715 was conducted in healthy volunteers, using a maximum dose of 60ug/kg of rhIL-7-hyFc. However, current clinical trials being conducted for cancer patients (NCT03901573, NCT04594811, NCT04332653) include doses ranging from 480ug/kg to 1200ug/kg, and no adverse events have been reported even at the highest dose. Therefore the 10 mg/kg dose can be converted to human clinical trials. We have added this information to the discussion in the updated manuscript as below:

"we used a dose of 10mg/kg rhIL-7-hyFc, to achieve maximum effect in these preclinical models. While it is likely that lower doses of rhIL-7-hyFc would be equally effective, the 10mg/kg dose in mice is equivalent to 813ug/kg in humans, and current clinical trials with rhIL-7-hyFc have used doses up to 1200ug/kg, without any serious adverse events²⁴⁻²⁶. Therefore, our preclinical dosing can potentially be directly converted to equivalent dosing in human subjects."

- *Figure 1H is difficult to interpret without a legend specifying the color coding. Please edit as appropriate.*

Response: We appreciate the reviewer's comment on this ambiguity in our figure. The color codes in Figure 1H reflect the numbers of cytokines secreted by the cells, as a measurement of polyfunctionality. We have updated the figure to clarify this.

- *Figure 1J: again difficult to distinguish visually major differences across the three groups. PCA or t-SNE analysis (or other dimensionality-reduction approaches) could be useful to show which analytes accounts for most of the variance in polyfunctionality.*

Response: The polyfunctional strength index (PSI) incorporates both polyfunctionality and also signal intensity into one measurement, and this is the main readout of this assay. For clarity we have moved both the polyfunctionality and the signal intensity measurements to Supplemental Figure 2, as these are redundant to the PSI.

- Regarding the statement p7 to overcome manufacturing issues with CAR-T cell target dose not being reached. I think this is weak rationale, the percentage of manufacturing failures across products is currently low, e.g., with Yescarta 97% of products have been reported "on-spec". Insufficient dose is only one of many causes of manufacturing failures, poor cell viability being another important cause. See <https://www.biopharma-reporter.com/Article/2019/04/01/Gilead-produces-97-on-spec-Yescarta> A better angle could be to use rhIL-7-hyFc as a CAR-T dose-sparing strategy to allow subsequent repeat infusions in patients.

Response: We appreciate the reviewer's insight in this regard. While rhIL-7-hyFc could be used to overcome manufacturing issues with CAR T cells, this was not the primary intent of these studies. Our goal is to improve outcomes of CAR T cell therapy for patients, and the main benefit of rhIL-7-hyFc would be to increase the potency of this treatment. We agree with the reviewer that it could also be used as a dose-sparing strategy in the future, but we think that CAR T cells still have a long way to go before we can start scaling back on this treatment. To avoid confusion in this regard, we have omitted the statement of overcoming manufacturing issues in the revised manuscript, and added the following points to our discussion:

“currently CAR T cells are an autologous therapy, and the potency of each patient's CAR T cells can be quite variable^{6,27}. While some patients' CAR T cells are effective against tumor by themselves, the majority of patients have less robust T cell function, and the addition of rhIL-7-hyFc may be able to achieve tumor regression in these individuals who would otherwise have succumbed to refractory disease. Additionally, if allogeneic CAR T cell products advance to the clinic in the future, rhIL-7-hyFc can be used to lower the minimum required dose of CAR T cells per patient, which could be a cost-effective strategy to overcome the financial toxicity of this therapy.”

- To demonstrate that rhIL-7-hyFc can improve persistence, beyond initial peak expansion, it would be useful to compare rhIL-7-hyFc-treated mice to vehicle-treated mice receiving an intermediate/high dose of CAR-T cells, with the goal of obtaining comparable CAR-T cell peak expansion, and to compare the CAR-T cell AUC or CAR-T numbers at selected timepoints.

Response: We thank the reviewer for this suggestion. We have performed the experiment proposed by the reviewer in our immunocompetent mouse model with mCART19, where limiting doses of CAR T cells were given with either vehicle or rhIL-7-hyFc. We find that indeed rhIL-7-hyFc can enhance both the peak expansion and the persistence of CAR T cells beyond that seen with CAR T cells alone at a five-fold higher dose. These findings have been added to the revised manuscript as **Figure 6C**, reproduced below:

Revised Figure 6C:

- Authors report p11 somewhat provocative data with decreased “genotypic” exhaustion (based on the author’s own scoring metric) across time. Can the authors put forth hypotheses and limitations to explain this?

Response: We thank the reviewer for giving us the opportunity to clarify the exhaustion score that we calculated from the scRNA-seq data on UCART19 after rhIL-7-hyFc treatment. The exhaustion scoring metric has been used in multiple prior scRNA-seq publications (PMID: 33795428, 32788748, 27124452, 28622514), and we have added these references to the revised manuscript. We used well-known T cell exhaustion markers to calculate the exhaustion score; however, to reassure the reviewer that this is not an arbitrary measure that was generated for our convenience, we also re-analyzed our data using the exhaustion and memory score devised by Weber et al, Science 2021, in which they demonstrated that excessive CAR signaling could induce T cell exhaustion (**Figure R1**). We continue to see a decrease in exhaustion and increase in memory phenotype with the presence of rhIL-7-hyFc using this independent method, as shown below:

- Figure 5: did the authors look at TOX and Ki-67 expression specifically?

Response: TOX2 and MKI67 expression were evaluated but not specifically reported on in the paper. We have included plots here for the reviewer (**Figure R2**). In general clusters with consistent high expression of MKI67 included clusters 2,3,13,8 and 5 across timepoints with other clusters showing variability based on timepoint. TOX2 was not as readily expressed but at Week 4 was present predominantly in Cluster 10. These results are consistent with our conclusions that rhIL-7-hyFc increases proliferation and decreases exhaustion in CAR T cells.

- What is the post-in vivo expansion phenotype of CAR T cells treated with higher dose without rhIL-7-hyFc? Differences compared to rhIL-7-hyFc-treated mice?

Response: Unfortunately, we have not been able to phenotype human CAR T cells in vivo without rhIL-7-hyFc treatment, as cell numbers were insufficient for detailed analysis. We did analyze murine CAR T cells with and without rhIL-7-hyFc, where we could also compare the CAR T cell phenotype to that of endogenous T cells, which both respond to rhIL-7-hyFc. We find that CAR T cells are predominantly effector memory, and these slightly increase with rhIL-7-hyFc, along with an increase in the proportion of effector cells, which leads to a commensurate decrease in central memory cells. This is in contrast to endogenous T cells, which are primarily naïve cells, and show a slight increase in central memory and effector memory populations after rhIL-7-hyFc. Given the increase in absolute numbers of CAR T cells with rhIL-7-hyFc, all CAR T

cell populations are numerically increased with rhIL-7-hyFc, but the highest fold-change is observed with the CD4+effector and effector memory cells. These findings have been added to the revised paper as **Figure 5E-F**, reproduced below:

Revised Figure 5E-F

- Looking at the data showed in Fig 5C, cluster 0 and 1 seem connected to multiple clusters, not only Cluster 4 as reported in the manuscript.

Response: The reviewer is correct in identifying there are multiple transitions from clusters 0 and 1. We have updated the text to reflect this point more clearly, as follows:

“A UMAP of all T cell clusters and the overlapping trajectory indicate that cells starting with cluster 4 (naïve T cells) transition to several downstream clusters, including cluster 0 and cluster 1, which harbor more of an effector memory state, before transitioning to a number of additional T cell states”

- Did the authors experience difficulties with xenoGVHD prompting them to knock-out the endogenous TCR? If so, this should be clearly mentioned and additional data should be provided as Supplementary material.

Response: We have not attempted these studies with TCR-intact human T cells, as xenogeneic GVHD is a well-known phenomenon when giving human T cells to NSG mice (PMID: 21572464, 22937164, 30383447). We have also previously shown that wild-type T cells induce GVHD in NSG mice, and this can be prevented by use of TCR-deleted T cells (Cooper et al, Leukemia 2018). Below we show the pertinent figure from this paper for the reviewer:

Editorial Note: Figure below reprinted by permission from Springer Nature Customer Service Centre GmbH: Springer Nature. Cooper, M.L., Choi, J., Staser, K. et al. An “off-the-shelf” fratricide-resistant CAR-T for the treatment of T cell hematologic malignancies. *Leukemia* **32**, 1970–1983 (2018). <https://doi.org/10.1038/s41375-018-0065-5>

- Discussion: I agree with the authors that immunogenicity may be worsened with rhIL-7-hyFc in the allogeneic setting. Importantly, this could also be a concern in the autologous setting, a situation in which CD8+-mediated anti-CAR immune responses have been detected (see Gauthier J et al Blood 2020). This should be mentioned in the discussion.

Response: We thank the reviewer for this important observation. We have added additional data to show that in immunocompetent mice, we do not see rejection of congenic mCART19, despite the CAR sequence and also presence of GFP. We have expanded upon these findings in the discussion as suggested by the reviewer, as follows:

“However, NSG mouse models fail to address the potential for host versus graft effect, specifically the concern that rhIL-7-hyFc may contribute to host rejection of CAR T cells, which can occur even in the autologous setting by immune responses directed against the CAR²⁸. Therefore, we also tested rhIL-7-hyFc in immunocompetent mouse models, using congenic CD45.1+ mCART19 linked to GFP. While rhIL-7-hyFc led to a predominant expansion of endogenous lymphocytes, this did not lead to rejection of CAR T cells, as mCART19 persisted for many weeks and also mediated prolonged B cell aplasia up to 100 days after initial T cell

injection. While the T cells themselves would not be foreign to the host, the presence of the synthetic CAR construct and GFP could theoretically lead to rejection, and so the persistence of CAR T cells in this setting was reassuring in that regard."

- *Methods: a section should describe the murine models in more details*

Response: We have re-written the methods to describe the murine models in greater detail.

- *Statistical testing was not always performed (e.g. in vitro assays in Fig 1). Was there a rationale to decide whether or not to perform hypothesis testing?*

Response: We updated the figures to include statistical testing on all experiments.

- *"Based on the law of diminishing returns, Mead recommended that a degree of freedom (DF) of 10-20 associated with error term in an ANOVA will be adequate for a pilot study to estimate preliminary information [42]."*
is this statement appropriate or copy-pasting error?

Response: This was an appropriate statement, however given the reviewer's concern we have revised this statement in the statistical methods to add clarity, as follows:

"Based on the law of diminishing returns, Mead recommended that a pilot study with a sample size of 10-20 subjects would be adequate to estimate preliminary information³⁹"

Jordan Gauthier, MD, MSc

REVIEWER COMMENTS

Reviewer #1 (Remarks to the Author):

The overall changes made to the manuscript text, figures and legends has improved the quality of the paper. We are satisfied with the improvements made and recommend publication.

Reviewer #2 (Remarks to the Author):

This is a resubmission of a manuscript illustrating the in vivo efficacy of modified IL-7 for the stimulation and expansion of CAR-T cells in the treatment of hematologic malignancies.

The manuscript significantly improved through the addition of new data and the basic concept is convincing. Yet, the functional relevance of analyzed T cell differentiation states (central vs. effector memory vs. naive etc) after in vitro and/or in vivo stimulation does not become entirely clear.

Minor comments:

Fig 1c: not necessary

Fig 1f: not really informative (even in combination with supplemental data)

Fig 5 e&f: limited scientific information, even in combination with scRNAseq data from Fig. 8. Link to functional relevance?

Fig. 6: With only 4 animals/group and the complexity of the model system this experiment seems premature for publication. A20 is often difficult to engraft after iv injection w/o prior irradiation. The better set-up for this experiment would have been to treat with cytoxan first, followed by the injection of A20 and 2-3d later CAR treatment. This way, the benefit of CAR therapy could probably be proven more convincingly.

Fig. 6c: it would be helpful to keep y-axis constant

Fig. 6d: the log difference in PB B cell numbers between treatment groups is neither explained in text nor legend. Is this the IL-7 effect on B cell reconstitution (IL7 only, middle panel)?

Reviewer #3 (Remarks to the Author):

Authors have done an outstanding job addressing all my comments.
I recommend acceptance.

REVIEWER COMMENTS

Reviewer #1 (Remarks to the Author):

The overall changes made to the manuscript text, figures and legends has improved the quality of the paper. We are satisfied with the improvements made and recommend publication.

Response: We thank the reviewer for their recommendation.

Reviewer #2 (Remarks to the Author):

This is a resubmission of a manuscript illustrating the in vivo efficacy of modified IL-7 for the stimulation and expansion of CAR-T cells in the treatment of hematologic malignancies.

The manuscript significantly improved through the addition of new data and the basic concept is convincing. Yet, the functional relevance of analyzed T cell differentiation states (central vs. effector memory vs. naive etc) after in vitro and/or in vivo stimulation does not become entirely clear.

Response: We respectfully disagree with the reviewer, as in our opinion the analysis of CAR T cell differentiation states adds an extra dimension to the paper that should be interesting to the readers. Contrary to the literature and our initial expectations, we see decreased central memory and increased effector memory CAR T cells in vivo after rhIL-7-hyFc treatment (Fig 5e-f, Fig 8, Supplemental Fig 5, and Supplemental Fig 9). We expended significant effort to ensure that this unexpected finding was not an artifact of the experimental system, but reproducible across different models (using both human and murine CAR T cells) and different methods (flow cytometry and scRNA-seq). We postulate that this effector memory predominance of CAR T cells may be due to the inherent bias of CAR T cells towards effector cells that are amplified by the presence of antigen-positive target cells providing additional activation and proliferation signals in vivo. In terms of functional relevance, long-term surviving mice after mCART19 and rhIL-7-hyFc demonstrated better protection against tumor rechallenge on day 100 (Fig 5g) and maintained B cell aplasia past 100 days (Fig 6e), indicating that rhIL-7-hyFc clearly enhances the immune protection provided by CAR T cells against target antigen, despite the absence of phenotypic Tcm generation. We have modified the discussion to highlight these findings in more detail as follows:

Lines 409-420:

“However, contrary to our expectations, both human and murine CAR T cells displayed a predominantly effector memory (Tem) phenotype after in vivo rhIL-7-hyFc treatment. In the murine model we were able to compare the response of endogenous T cells and CAR T cells to rhIL-7-hyFc. For endogenous T cells, the effects of rhIL-7-hyFc were approximately equivalent between CD4+ and CD8+ T cells, and led to preferential increase of memory T cells. However, in CAR T cells the effects of rhIL-7-hyFc were most prominent in the CD4+ effector cells. These differences may be due to the inherent bias of CAR T cells towards effector cells that are amplified by the presence of antigen-positive target cells providing additional activation and proliferation signals in vivo. Functional assessment of CAR T cells after rhIL-7-hyFc in vivo demonstrated better protection against tumor rechallenge and prolonged B cell aplasia as

compared to CAR T cells alone, indicating that rhIL-7-hyFc does enhance immune protection, despite the absence of phenotypic Tcm generation.”

Minor comments:

Fig 1c: not necessary

Response: We agree with the reviewer that Fig 1c is not absolutely necessary, as the experimental setup is described in the figure legend and manuscript text. However, many readers scan through the figures of a paper quickly, and having a schema can help the readers quickly grasp the meaning of the data presented. We do not see any drawback to keeping this figure.

Fig 1f: not really informative (even in combination with supplemental data)

Response: We respectfully disagree with the reviewer, as this figure shows that the polyfunctionality of the CAR T cells after rhIL-7-hyFc expansion is preserved as compared to day 0 input T cells. One of the major concerns we initially had when observing the impressive expansion of CAR T cells with rhIL-7-hyFc was that the quality of CAR T cells might be compromised after so many cell divisions, and this data provides preliminary evidence that this is not the case in vitro. Our subsequent data expands upon this and shows that the quality of CAR T cells is actually enhanced after in vivo rhIL-7-hyFc treatment. Again, we do not see any downside to keeping this figure, and also the supporting data in Supplemental Fig 2, in the manuscript.

Fig 5 e&f: limited scientific information, even in combination with scRNAseq data from Fig. 8. Link to functional relevance?

Response: We appreciate the opportunity to clarify the relevance of our data in Fig 5e and f. In Figure 8 and Supplemental Fig 5 we show that human UCART19 treated with rhIL-7-hyFc in immunodeficient NSG mice retain a predominantly effector memory phenotype. Fig 5e & f corroborates these findings for mCART19 treated with rhIL-7-hyFc in Balb/c mice, providing orthogonal validation of these findings in an entirely different immunocompetent mouse model. Additionally, we are able to compare the expansion of endogenous T cells, which are mostly naïve, to CAR T cells, which are more skewed towards memory T cells. We see an overall better expansion of CAR T cells across all subtypes at this timepoint as compared to endogenous T cells, and better expansion of CD4+ effector T cells with the presence of the CAR. These differences may be due to the inherent bias of CAR T cells towards effector cells that are amplified by the presence of antigen-positive target cells providing additional activation and proliferation signals in vivo. The functional relevance is shown in Figure 5g and h, where rhIL-7-hyFc treatment after CAR T cells leads to better protection against tumor rechallenge. We have added additional text to clarify these findings in the results and the discussion of the manuscript as follows:

Lines 209-218:

“Additional evaluation of T cell phenotype on day +57 by flow cytometry revealed that in contrast to the endogenous T cells, which were predominantly naïve T cells, mCART19 cells were mostly comprised of memory T cells (Figure 5E, Supplemental Figure 8-9). Addition of rhIL-7-hyFc did not significantly change the proportion of T cell subsets for the most part, although there was a slight increase in effector CAR T cells and decrease in central memory CAR T cells. In general we observed better expansion of CAR T cells as compared to endogenous T cells at this later timepoint, with the largest fold-change seen in the CD4+ effector and effector memory CAR T cells after rhIL-7-hyFc (Figure 5F). These findings corroborate our findings in human CAR T cells, in that rhIL-7-hyFc leads to increased CAR T cell numbers, and this is more prominent within the effector and effector memory cell populations.

Lines 411-420:

“In the murine model we were able to compare the response of endogenous T cells and CAR T cells to rhIL-7-hyFc. For endogenous T cells, the effects of rhIL-7-hyFc were approximately equivalent between CD4+ and CD8+ T cells, and led to preferential increased of memory T cells. However, in CAR T cells the effects of rhIL-7-hyFc were most prominent in the CD4+ effector cells. These differences may be due to the inherent bias of CAR T cells towards effector cells that are amplified by the presence of antigen-positive target cells providing additional activation and proliferation signals in vivo.”

Fig. 6: With only 4 animals/group and the complexity of the model system this experiment seems premature for publication. A20 is often difficult to engraft after iv injection w/o prior irradiation. The better set-up for this experiment would have been to treat with cytoxan first, followed by the injection of A20 and 2-3d later CAR treatment. This way, the benefit of CAR therapy could probably be proven more convincingly.

Response: We agree with the reviewer that the experimental setup was not optimal in this case, as the majority of mice did not develop disease from the A20 tumor cells, which we attribute to the cytotoxic effect of Cytoxan. We would hesitate to perform the experiment as the reviewer suggests, however, as we usually see rapid immune reconstitution after Cytoxan, and so a 2-3 day delay between Cytoxan and CAR T cells may lead to poor engraftment of the T cells. Our

solution was to irradiate the mice prior to injection with A20 and mCART19, as this leads to more profound and long-lasting immunosuppression, and we show the benefit of rhIL-7-hyFc on CAR therapy in this model in Figure 5.

The drawback of using radiation is that the mice develop prolonged thrombocytopenia causing hemorrhage and death after bleeding, so we cannot monitor CAR T cell or B cell numbers during the first few weeks. Therefore, Fig 6 is still informative, as we show marked differences in CAR T cell expansion and B cell aplasia after different doses of mCART19 with and without rhIL-7-hyFc. These differences are quite profound even with the few numbers of mice. Additionally, the data is consistent with both murine CAR T cells in Balb/c and human CAR T cells in NSG mice, which gives us greater confidence in our conclusions that rhIL-7-hyFc can compensate for suboptimal CAR T cell numbers.

Fig. 6c: it would be helpful to keep y-axis constant

Response: We thank the reviewer for this helpful comment on the visualization of our data. We have modified the y-axis of the 2.5e5 cells (middle panel) to match the axis of the 1.25e6 cells (right panel). For the 5e4 cells (left panel), matching the y-axis to the other graphs would make the values undecipherable, so we have slightly modified the y-axis to maximum 300 (rather than 150, as originally shown), and noted in the legend that the axis is on a different scale. These changes are reproduced below:

Revised Fig 6C:

Fig. 6d: the log difference in PB B cell numbers between treatment groups is neither explained in text nor legend. Is this the IL-7 effect on B cell reconstitution (IL7 only, middle panel)?

Response: We thank the reviewer for giving us the opportunity to clarify the increase in B cell numbers after rhIL-7-hyFc in our immunocompetent mouse models. We do see a consistent increase in circulating B cells after rhIL-7-hyFc administration (this is also shown in Figure 4f with the C57BL/6 mice), and this is a known effect of IL-7 on murine B cells (ref PMID: 1999226, 8207207). We did not elaborate on this in detail as these findings are unique to mice, and in humans IL-7 leads to expansion of bone marrow early B cell progenitors but not circulating mature B cells (ref PMID: 20068111). However, in Figure 6 these findings are particularly interesting as rhIL-7-hyFc is stimulating both the endogenous target B cells and the effector CAR T cells in vivo, and as the reviewer notes the fold-expansion is dramatically higher for the B cells in the control mice than for CAR T cells in the treated mice. Despite this, mice treated

with mCART19+rhIL-7-hyFc still develop B cell aplasia at all but the lowest dose level, and even with 5×10^4 mCART19 cells we observe transiently less B cells as compared to control mice. These findings indicate that the effect of rhIL-7-hyFc is more potent for CAR T cells than for B cells, and strengthens our argument that the effects of rhIL-7-hyFc on CAR T cells extend beyond quantitative increase in numbers, and that it also improves the quality of CAR T cells. We have added additional text to clarify these findings in the results section as follows:

Line 238-245:

“Using circulating B cells as a biomarker of CAR T cell activity, we found that without rhIL-7-hyFc, mice treated with 2.5×10^5 mCART19 recovered B cells after 3 weeks, while addition of rhIL-7-hyFc led to prolonged B cell aplasia. **Notably, control mice had massive increase in B cell numbers after rhIL-7-hyFc, but mice treated with 2.5×10^5 or more mCART19 with rhIL-7-hyFc were still able to completely eliminate all detectable circulating B cells.** Also, while the 5×10^4 mCART19 + rhIL-7-hyFc treated mice were not able to clear B cells at any time point, these mice did not exhibit the marked increase in B cell numbers seen in the control group, suggesting that even this low cell dose had some effect against target cells when augmented by rhIL-7-hyFc”

Reviewer #3 (Remarks to the Author):

Authors have done an outstanding job addressing all my comments.
I recommend acceptance.

Response: We thank the reviewer for their recommendation.